# A novel role for CSA in the regulation of nuclear envelope integrity: uncovering a non-canonical function

Denny Yang[1,2], Austin Lai[3], Amelie Davies[1] , Anne FJ Janssen[1,3] , Matthew O Ellis[2], Delphine Larrieu[1,3]

**Cockayne syndrome (CS) is a premature ageing condition characterized by microcephaly, growth failure, and neurodegeneration. It is caused by mutations in *ERCC6* or *ERCC8* encoding for Cockayne syndrome B (CSB) and A (CSA) proteins, respectively. CSA and CSB have well-characterized roles in transcription-coupled nucleotide excision repair, responsible for removing bulky DNA lesions, including those caused by UV irradiation. Here, we report that CSA dysfunction causes defects in the nuclear envelope (NE) integrity. NE dysfunction is characteristic of progeroid disorders caused by a mutation in NE proteins, such as Hutchinson–Gilford progeria syndrome. However, it has never been reported in Cockayne syndrome. We observed CSA dysfunction affected LEMD2 incorporation at the NE and increased actin stress fibers that contributed to enhanced mechanical stress to the NE. Altogether, these led to NE abnormalities associated with the activation of the cGAS/STING pathway. Targeting the linker of the nucleoskeleton and cytoskeleton complex was sufficient to rescue these phenotypes. This work reveals NE dysfunction in a progeroid syndrome caused by mutations in a DNA damage repair protein, reinforcing the connection between NE deregulation and ageing.**

## Introduction

Progeroid disorders are a group of incurable rare diseases that resemble many aspects of physiological ageing (1, 2). Clinical manifestations include premature and accelerated ageing, developmental delay, hair loss, vision and hearing loss, progressive neurodegeneration, cardiovascular diseases, osteoporosis, and early death (1). Progeroid disorders can be characterized by two distinct subtypes at the molecular level. The conditions caused by mutations in genes involved in the DNA damage response and repair, including Werner syndrome, ataxia telangiectasia, Bloom syndrome, Cockayne syndrome and xeroderma pigmentosum, and the conditions caused by mutations in nuclear envelope (NE) genes,

such as Hutchinson–Gilford progeria syndrome (HGPS) or Nestor–Guillermo progeria syndrome (3).

Cockayne syndrome (CS), is an autosomal recessive condition caused by mutations in either *ERCC8* (20% of cases) or *ERCC6* (80% of cases) genes, encoding for CSA and CSB proteins, respectively (4, 5, 6). CS occurs at a rate of 1 in 300,000–500,000 live births in the USA and Europe, and the patients have a life expectancy ranging from 5 to 16 yr (7, 8). CS is mainly characterized by growth failure and neurological abnormalities (9). Other clinical manifestations include cataracts, microcephaly, and cutaneous photosensitivity (10). The main described function for CSA and CSB proteins is in transcription-coupled nucleotide excision repair (TC-NER). TC-NER is a DNA damage repair mechanism that removes bulky DNA adducts such as those induced by UV, mainly including 6-4 pyrimidine-pyrimidine (6-4 PP) photoproducts and cyclobutane-pyrimidine dimers (11, 12). The loss-of-function mutations in *ERCC8* and *ERCC6* genes occurring in Cockayne syndrome patients prevent the removal of these bulky DNA lesions by TC-NER (12, 13). This results in the progressive accumulation of DNA damage in CS patient cells, and explains the cellular photosensitivity in CS patients (5, 14). However, other clinical manifestations of CS such as neurodegeneration (15) cannot be explained by defects in TC-NER because other TC-NER-associated syndromes including UV-sensitive syndrome do not display neurodegeneration (3, 16). This suggests that there are additional mechanisms underlying tissue dysfunction in CS patients. New functions for CS proteins in mitochondrial autophagy, oxidative stress response (17, 18), and transcriptional regulation have been suggested to contribute to these phenotypes (15).

The NE plays a pivotal role in regulating cellular homeostasis, maintaining the structural architecture of the nucleus, and controlling chromatin organization, nucleocytoplasmic transport, and propagating mechanical cues from the cytoplasm to the nucleus (19, 20, 21). The NE is composed of an inner nuclear membrane (INM) and an outer nuclear membrane (ONM) which are both two lipid bilayers. The ONM is continuous with the ER and merges with the INM at the nuclear pore complexes. The nuclear lamina is a protein meshwork composed of intermediate filaments of A-type lamin

[1]Department of Pharmacology, University of Cambridge, Cambridge, UK   [2]UK Dementia Research Institute, Island Research Building, Cambridge, UK   [3]Cambridge Institute for Medical Research, The Keith Peters Building, Cambridge, UK

Correspondence: dlarrieu@altoslabs.com
Austin Lai's present address is Duke University Medical Center, Durham, NC, USA
Delphine Larrieu's present address is Altos Labs, Cambridge Institute of Science, Cambridge, UK

(lamin A and C) and B-type lamin (lamin B1 and B2) proteins ([22]). Lamins interact with the chromatin and with other NE proteins including LAP2-emerin-MAN1 (LEM)-domain proteins ([23]). Extra-cellular mechanical signals are sensed and propagated to the nucleus through the linker of the nucleoskeleton and cytoskeleton (LINC) complex ([24]), composed of Sad1/UNC-84 (SUN)-domain proteins (SUN1 and SUN2) and Klarsicht/ANC-1/Syne-1 homology (KASH)-domain proteins (Nesprin1–4) ([25]). The LINC complex spans over the INM and ONM connecting cytoskeletal components to the nucleus. Various stresses can trigger loss of NE integrity. These include deleterious mutations in genes encoding for NE proteins (e.g., *LMNA* or *LMNB1*) ([26]) or mechanical stress (e.g., cancer cells migrating through tiny blood vessels during metastasis or tissues subjected to contractions such as skeletal or cardiac muscles) ([27], [28], [29]). These stresses enhance the probability for cells to experience NE ruptures during interphase. These ruptures are typically preceded by the formation of gaps in the lamina that generate a weak point at the NE. This leads to the formation of nuclear blebs in which the chromatin can protrude through the lamina gaps ([30], [31]). Nuclear blebs can then rupture if the mechanical pressure is not resolved. This results in exchange of content between the nucleus and the cytoplasm which can result in DNA damage and activation of the innate immune cGAS/STING pathway, which is associated with inflammation ([32]). The NE repair process is mediated by barrier-to-autointegration factor 1 (BAF) where LEM domain proteins, A-type lamins and the endosomal sorting complexes required for transport-III protein complex are recruited to the rupture site to facilitate the resealing of the nuclear membrane and to restore NE integrity ([31], [33], [34], [35]).

Here, we found that CSA KO or loss-of-function mutation was associated with multiple NE abnormalities, causing ruptures and activation of the cGAS/STING innate immune pathway. More specifically, we revealed two mechanisms that drive NE defects in CSA patient cells: ([1]) the decreased formation of LEMD2-lamin A/C complexes at the NE and ([2]) the increased actin stress fibers that generate mechanical tension on the NE. This work sheds light on a new role for CSA in NE regulation and suggests a new mechanism that can contribute to the loss of homeostasis in CS-A cells.

## Results

NE defects are characteristic of premature ageing conditions associated with mutations in NE proteins, such as HGPS, restrictive dermopathy, atypical progeria syndrome, and Nestor–Guillermo progeria syndrome ([3]). In addition, there is now mounting evidence that NE dysfunction can also occur in physiological ageing ([36]). To further investigate the connection between NE dysfunction and ageing phenotypes, we assessed NE integrity in Cockayne syndrome (CS), a premature ageing condition not caused by mutations in NE-associated proteins. To mimic the loss-of-function mutations observed in CS, we used HAP1 cells in which *ERCC8* (CSA) or *ERCC6* (CSB) were knocked out. Interestingly, we observed that knocking out *CSA* but not *CSB* in these cells induced aberrant nuclear morphology ([Fig 1A]). To characterize the NE phenotypes further, we

obtained CS patient-derived cell lines carrying loss-of-function mutations causing destabilization of CSA and CSB in CSA (CS-A cells) or CSB (CS-B cells), respectively ([6]), and their respective isogenic control cell lines WT(HA-CSA) and WT(HA-GFP-CSB) ([37]). The expression of HA-CSA and HA-GFP-CSB in the isogenic control cell lines and the absence of CSA and CSB expression in CS-A and CS-B cells, respectively, were confirmed by immunoblotting ([Fig 1B]). We first assessed the nuclear shape and the presence of nuclear blebs. Nuclear blebs are a reliable proxy to assess NE ruptures as blebs occur where the NE is weakened and often result in NE ruptures ([30], [38]). CS-A cells showed misshapen nuclei with a significantly lower nuclear circularity (represented by the reduced form factor) and a higher percentage of blebbing compared with the WT(HA-CSA) ([Fig 1C and D]). Conversely, CS-B cells showed a slight increase in nuclear circularity and no difference in the percentage of blebbing compared with WT(HA-GFP-CSB) cells ([Fig 1E and F]). As a more direct indication of NE rupture, we quantified the percentage of nuclei with cyclic GMP-AMP synthase (cGAS) foci. cGAS is a double-stranded DNA sensor that binds to cytosolic DNA, and upon NE rupture, cGAS binds to the genomic DNA being exposed at the site of NE rupture ([39]). Consistent with the increased blebbing, we observed that CS-A cells displayed more cGAS foci per nucleus compared with WT(HA-CSA) ([Fig 1G and H]), whereas no significant differences were detected between CS-B and WT(HA-GFP-CSB) cells ([Fig 1I and J]). These findings were validated by knocking out *CSA* in an additional human fibroblast cell line (AG10803) using CRISPR/Cas9 ([Fig S1A]). Again, *CSA* KO caused a significant reduction in nuclear circularity, increased nuclear blebbing, and nuclear cGAS foci accumulation ([Fig S1B–F]). Together, these observations showed that CSA prevents NE deformation, blebbing, and ruptures.

To explore whether the loss of CSA may cause NE defects by affecting the expression or localization of NE proteins, we probed a panel of "core" NE proteins including lamin A, lamin C, lamin B1, LEMD2, emerin, and SUN1 by Western blotting ([Fig 2A]). There was no noticeable difference in the expression of these proteins between WT(HA-CSA) and CS-A cells. We then assessed the subcellular localization of these NE proteins using immunofluorescence. Similarly, we did not see any difference between the WT(HA-CSA) and CS-A cells ([Fig 2B]). For LEMD2, because of the high background signal given by the antibody, it was challenging to visualize its NE localization. Therefore, we performed pre-extraction to visualize the "insoluble" pool of LEMD2 ([Fig 2C]). The specificity of the detected LEMD2 signal after pre-extraction was confirmed using a LEMD2 siRNA ([Fig S2]). We showed that the amount of insoluble LEMD2 was significantly lower (~20%) in CS-A cells compared with WT(HA-CSA) ([Fig 2D]) and in CSA KO AG10803 cells ([Fig S3A and B]). This result was confirmed by Western blotting after a similar pre-extraction, whereas the LEMD2 expression level in the total cell lysate was unchanged ([Fig 2E and F]).

LEMD2 is a known A-type lamin-binding protein and previous literature suggests that the binding of these two proteins is important for maintaining the structural integrity of the nucleus ([40]). Because we observed a decrease in LEMD2 at the NE in CS-A cells, we speculated that the number of LEMD2-Lamin A/C complexes at the NE might also be reduced, which may be contributing to NE fragility. Using a Proximity Ligation Assay (PLA), we showed a

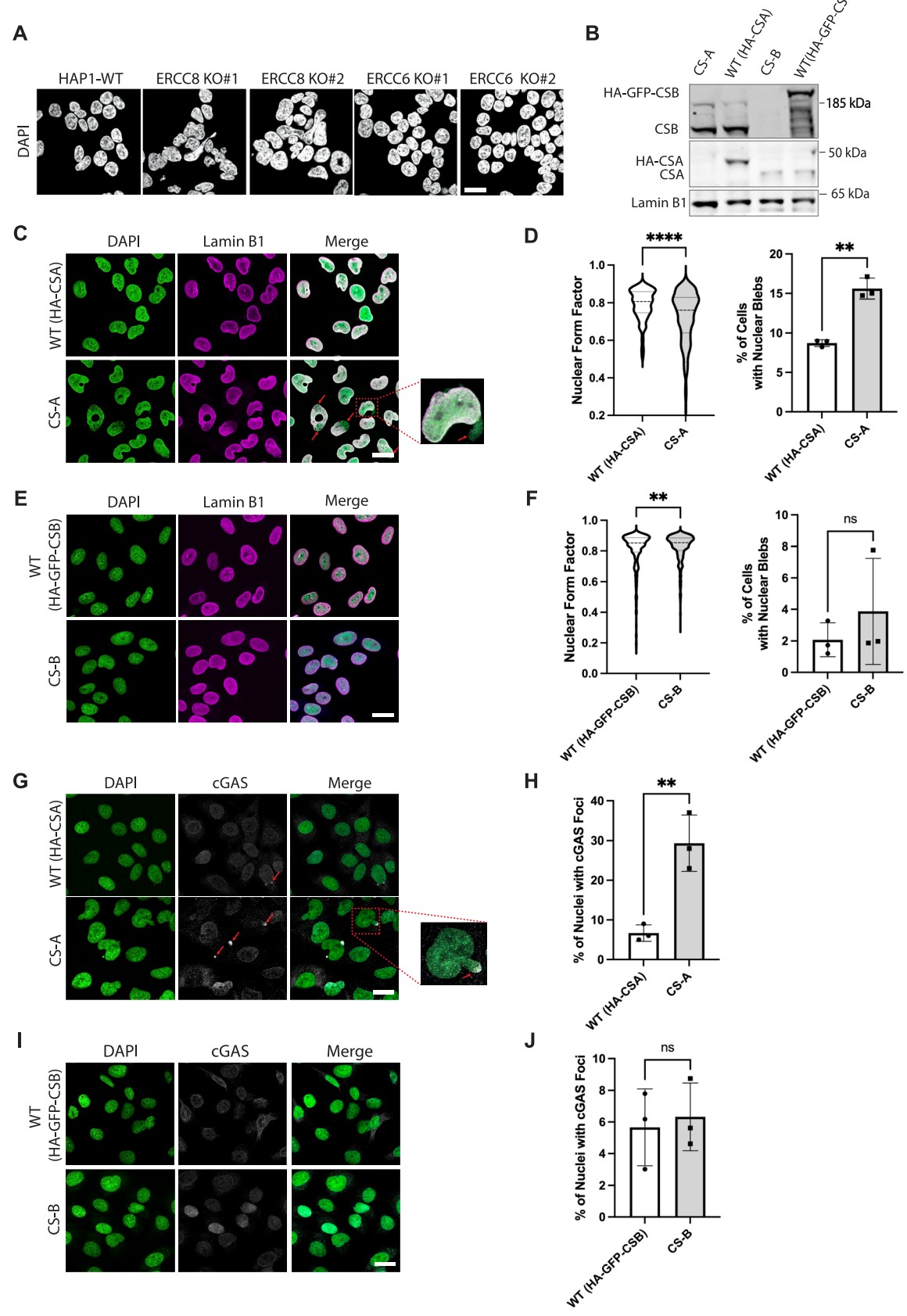

significant reduction in the number of PLA foci in CS-A cells compared with the WT(HA-CSA) cells, reflecting a reduced number of LEMD2-lamin A/C complexes at the level of individual nuclei (Fig 3A and B). To assess whether re-introducing LEMD2 could rescue NE defects in CSA-depleted cells, LEMD2-GFP was transiently overexpressed in WT(HA-CSA) and CS-A cells. We showed that CS-A cells transfected with exogenous LEMD2-GFP displayed significantly higher nuclear roundness and less nuclear blebbing (Fig 3C–E). Altogether, these results further supported the hypothesis that the reduction in LEMD2 at the NE contributes to the NE defects observed in CSA-depleted cells, To understand the potential mechanism by which the NE incorporation of LEMD2 might be affected in CSA-depleted cells, we investigated whether LEMD2 and CSA might be interacting with each other. Using BioID approaches, previous literature showed that LEMD2 is in proximity with known interactors of CSA, including CUL4A, DDB1, and CSN (41). Therefore, we hypothesized that LEMD2 could also interact with CSA and this interaction may be important to properly incorporate LEMD2 into the NE. To address this, we overexpressed LEMD2-GFP and Flag-CSA constructs, followed by GFP pulldown in WT(HA-CSA) cells. We observed that LEMD2-GFP co-immunoprecipitated with HA-CSA and Flag-CSA (Fig 4A). BAF was used as a positive control for known LEMD2 interactors (31, 35). This finding suggested that CSA can indeed interact with LEMD2.

To establish whether this interaction may be important in stabilizing or immobilizing LEMD2 at the NE, we performed FRAP experiments in WT(HA-CSA) and CS-A cells transiently expressing LEMD2-GFP (Fig 4B). Both WT(HA-CSA) and CS-A cells showed similar LEMD2-GFP recovery half-time and percentage of immobile LEMD2-GFP fraction (Fig 4C and D). Altogether, these data suggest that CSA can bind LEMD2 and that the absence of CSA in CS-A patient cells does not affect the mobility of LEMD2 at the NE but instead decreases insoluble levels of LEMD2 at the NE.

In searching for other mechanisms that could contribute to the loss of NE integrity in CS-A cells, we analyzed the publicly available bulk RNA sequencing (RNA-seq) dataset (GSE87540) obtained from WT(HA-CSA) and CS-A cells (42). Through DE gene analysis, we identified genes that were significantly up-regulated and down-regulated in CS-A cells compared with WT(HA-CSA) (Supplemental Data 1 and Supplemental Data 2). Through DE gene analysis, we found that genes involved in ER stress were differentially expressed (Fig 5A–C), which could be a consequence of the loss of proteostasis in CS-A cells as described previously (37, 43). More surprisingly, GO

enrichment analyzes identified differential transcript expressions of genes involved in cytoskeleton protein polymerization, including F-actin, $\alpha$-tubulin, and vimentin (Fig 5A–C). Because the cytoskeleton has a well-established role in regulating nuclear mechanical properties through binding to the LINC complex (44), we analyzed F-actin, $\alpha$-tubulin, and vimentin by immunofluorescence in CS-A cells (Fig 5D). There were no obvious changes in the $\alpha$-tubulin and the vimentin network organization between the WT(HA-CSA) and CS-A cells. However, we observed an increase in the formation of actin stress fibers in CS-A cells compared with WT(HA-CSA) (Fig 5D) and in the CSA KO AG10803 cell line (Fig S3C). Actin stress fibers are contractile molecular bundles in non-muscle cells that consist of parallel actin and myosin II filaments, and other actin-binding proteins including filamins, fascins, and actinins (45). We therefore decided to further pursue how mechanical forces generated by actin stress fibers may contribute to the loss of NE integrity in CS-A cells.

To this aim, we disrupted actin polymerization using cytochalasin D, a potent inhibitor of actin polymerization (Fig 6A). Interestingly, cytochalasin D treatment significantly improved nuclear circularity, decreased the number of nuclear blebs and reduced nuclear cGAS foci in CS-A cells, indicating an improvement in the NE integrity (Fig 6B–F). We then sought to perform a reverse experiment by treating WT(HA-CSA) and CS-A cells with Jasplakinolide (Fig 6A), an actin polymerization-inducing drug that stimulates the nucleation of actin filaments. We hypothesized that by stabilizing actin stress fibers in WT(HA-CSA) cells with Jasplakinolide, the Jasplakinolide-treated WT(HA-CSA) cells would display NE defects similar to what we observed in untreated CS-A cells. Consistent with our hypothesis, WT(HA-CSA) cells treated with Jasplakinolide displayed a significant reduction in nuclear circularity, increased nuclear blebbing, and increased nuclear cGAS foci (Fig 6B–F). When treating CS-A cells with Jasplakinolide, no significant change in either the nuclear circularity, percentage of nuclear blebs, or percentage of nuclei with cGAS foci were detected. This suggested that enhancing actin polymerization or stabilization cannot further increase the NE defects already present in CS-A cells. Altogether, these data suggested that the transcriptional dysregulation of genes affecting actin polymerization in CS-A cells contributes to the loss of NE integrity.

Cytoskeleton stress fibers can transduce mechanical forces to the NE through the LINC complex, consisting of the KASH-domain and SUN-domain proteins (25, 44). Therefore, we asked whether

**Figure 1. Loss of CSA but not CSB induces NE defects.**
**(A)** Representative DAPI staining images showing the effect of *CSA* or *CSB* KO on nuclear shape in the HAP1 cell line (representative images of two independent CRISPR-KO clones). Scale bar: 25 $\mu$m. **(B)** Immunoblot showing the expression of CSA and CSB in the indicated cell lines. **(C)** Representative DAPI and lamin B1 immunofluorescence images of WT(HA-CSA) and CS-A cells; scale bar: 25 $\mu$m. Red arrows indicate nuclear blebs and zoom-in image shows an example of nuclear bleb. **(D)** Quantification of nuclear form factor and percentage of blebbing for WT(HA-CSA) and CS-A cells, respectively, using a two-tailed unpaired t test (**$P < 0.01$, ****$P < 0.0001$), in n = 3 with >300 cells per experiment. **(E)** Representative DAPI and lamin B1 immunofluorescence images of WT(HA-GFP-CSB) and CS-B cells; scale bar: 25 $\mu$m. **(F)** Quantification of nuclear form factor and percentage of blebbing for WT(HA-GFP-CSB) and CS-B cells, respectively, using a two-tailed unpaired t test (ns $P > 0.05$, **$P < 0.01$), n = 3 with >300 cells per experiment. **(G)** Representative DAPI and cGAS immunofluorescence images of WT(HA-CSA) and CS-A cells; scale bar: 25 $\mu$m. Red arrows indicate cGAS foci and zoom-in image shows an example of cGAS foci. **(H)** Quantification of the percentage of nuclei with cGAS foci for WT(HA-CSA) and CS-A cells, respectively, using a two-tailed unpaired t test (**$P < 0.01$), n = 3 with >300 cells per experiment. **(I)** Representative DAPI and cGAS immunofluorescence images of WT(HA-GFP-CSB) and CS-B cells; scale bar: 25 $\mu$m. **(J)** Quantification of the percentage of nuclei with cGAS foci for WT(HA-GFP-CSB) and CS-B cells, respectively, using a two-tailed unpaired t test (ns $P > 0.05$), n = 3 with >300 cells per experiment. All experiments in this figure were n = 3 independent experiments.
Source data are available for this figure.

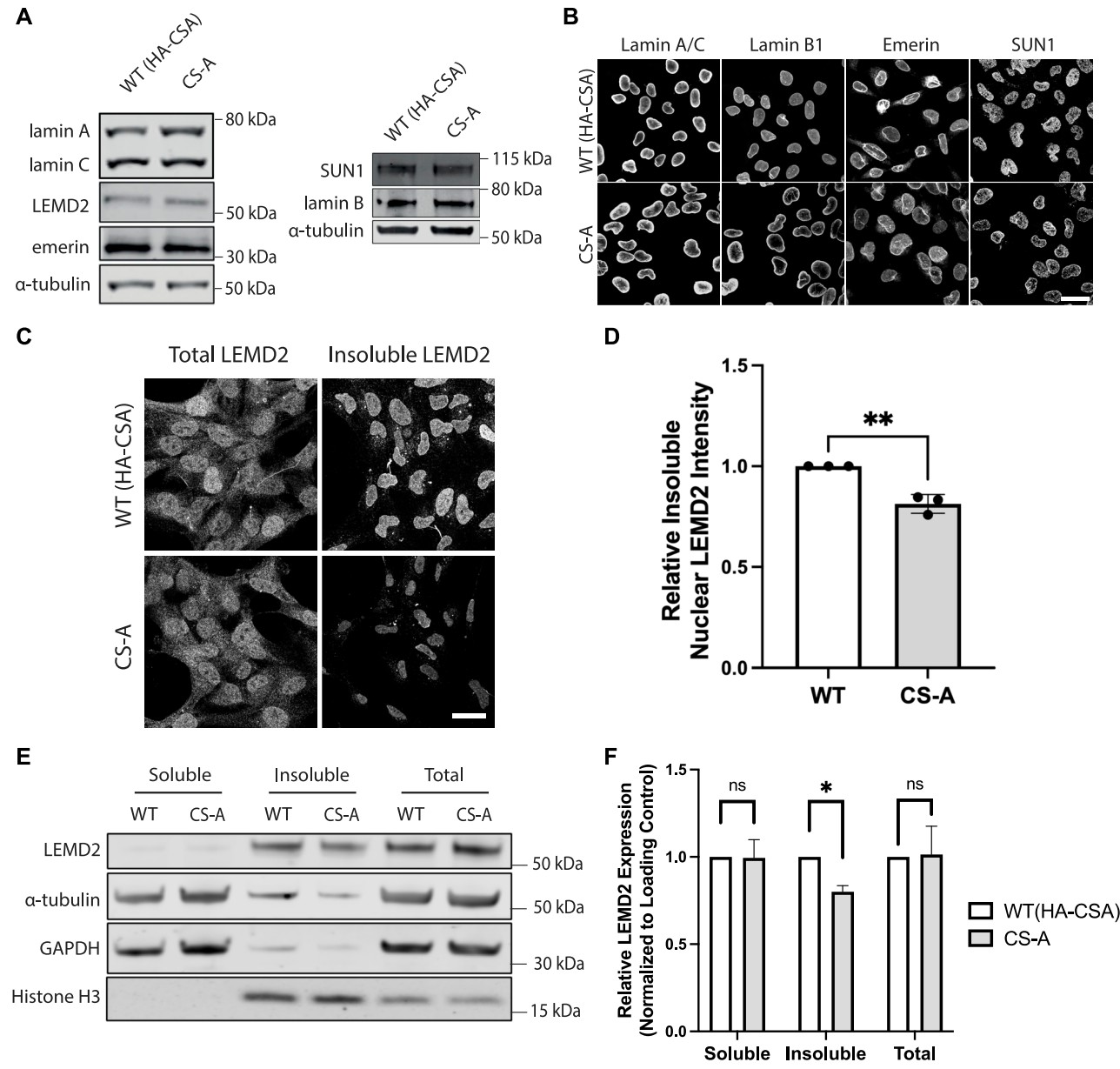

**Figure 2.  LEMD2 incorporation into the NE is decreased in CS-A cells.**
**(A)** Representative Western blot of the indicated NE proteins in WT(HA-CSA) and CS-A cells. **(B)** Representative immunofluorescence images of lamin A/C, lamin B1, emerin, and SUN1 staining; scale bar: 25 μm. **(C)** Representative immunofluorescence staining showing total and insoluble LEMD2 protein in WT(HA-CSA) and CS-A cells; scale bar: 25 μm. **(D)** Quantification of the insoluble LEMD2 showing a significant reduction in CS-A compared with WT(HA-CSA) using a two-tailed unpaired t test (**P < 0.01), n = 3 with >100 cells per experiment. **(E)** Representative immunoblot showing LEMD2 expression level in the soluble, insoluble, and whole cell extracts in WT(HA-CSA) and CS-A cells. **(E, F)** Quantification of the relative LEMD2 expression level from the western blots as shown in (E); normalized to GAPDH, histone H3, and α-tubulin, respectively. P-values were calculated using a two-tailed paired t tests (ns P > 0.05, *P < 0.05). All experiments in this figure were n = 3 independent experiments. Source data are available for this figure.

disrupting the cytoskeletal-to-NE interaction in CS-A cells by depleting components of the LINC complex could relieve the mechanical forces at the NE generated by actin stress fibers and improve the NE phenotypes. Previous studies showed that SUN1, but not SUN2, is the main interactor of KASH-domain proteins in the LINC complex assembly and depleting SUN1 inhibits NE rupture in cancer cell lines (46). For this reason, we depleted SUN1 using siRNA in WT(HA-CSA) and CS-A cells (Fig 7A). Immunofluorescence staining

of DAPI and lamin B1 was then performed in siSUN1-depleted WT(HA-CSA) and CS-A cells (Fig 7B) to identify nuclear blebs (38, 47). We observed that the depletion of SUN1 in CS-A cells significantly increased nuclear circularity and reduced the number of nuclear blebs and nuclear cGAS foci (Fig 7B–D, H, and I). The same was observed in *CSA* KO AG10804 cell line (Figs S1B–F and S4A). Interestingly, depleting other components of the LINC complex including SUN2 and Nesprin1 using siRNA in both CS-A cells and *CSA*

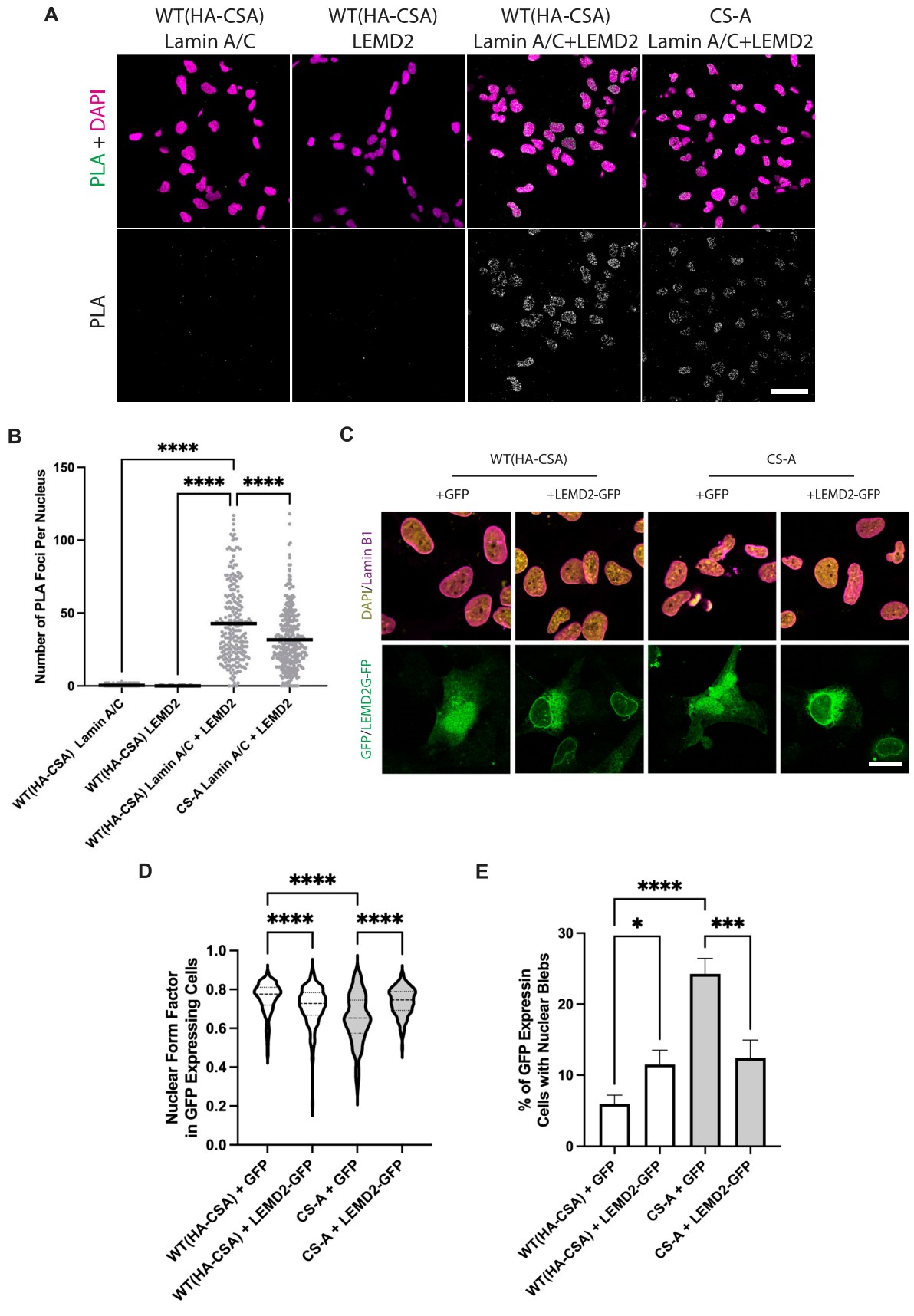

KO cells (Fig S4B–E), did not have the same effect and were not able to rescue the phenotypes including nuclear circularity, nuclear bleb, and nuclear cGAS foci (Figs 7E–I and S1B–F). This finding suggested that SUN1 specifically mediates NE deformation and rupture in CS-A cells, possibly through transmitting increased mechanical forces generated by actin stress fibers to the NE.

As CS-A cells showed an accumulation of cGAS foci (Fig 1G and H), we also wanted to assess whether this was associated with activating the cGAS/STING pathway upon NE rupture. Upon cGAS binding to DNA, cGAS can activate and trigger the phosphorylation of downstream effectors including TANK-binding kinase 1 (TBK1) and stimulator of interferon genes (STING) (32, 48). Indeed, we observed increased phosphorylation levels of TBK1 and STING in CS-A cells (Fig 8A and B) and in *CSA* KO AG10803 cells (Fig S5). Preventing the NE stress in CS-A cells and in *CSA* KO AG10803 cells by depleting SUN1 was able to reduce the cGAS/STING pathway activation (Figs 8C and D and S5A and B). Together, our results suggested that increased mechanical forces generated by the cytoskeleton contribute to NE defects in CS-A cells, causing ruptures leading to the cGAS/STING pathway activation which can be ameliorated by disrupting SUN1.

One of the best-established characteristics of CS-A cells is their sensitivity to DNA damage generated by UV irradiation (5) because of the role of CSA in TC-NER. Therefore, we wondered whether the function of CSA in NE regulation was related to its known function in DNA damage repair. To address this question, we set up a survival assay in response to the DNA damage induced by the UV-mimetic chemical 4NQO. As expected, the loss of CSA resulted in a significant reduction in cell survival after exposure to 4NQO, compared with the WT(HA-CSA) cells (Fig 9A). Interestingly, SUN1 depletion was not able to improve the survival of CS-A cells in response to 4NQO (Fig 9A and B). This experiment showed that even when NE defects and ruptures were rescued through SUN1 depletion, the CS-A cells were still sensitive to DNA damage. This result suggests that the function of CSA in maintaining the NE integrity is independent of its known function in TC-NER.

## Discussion

In this work, we report for the first time that the loss of function of CSA in CS-A patient cells leads to NE defects. Loss of NE integrity is a well-described feature of cells from HGPS patients and other laminopathies associated with NE dysfunction (49). This is reflected by the presence of nuclear deformation, NE ruptures, and lamin invaginations (50). More recently, NE defects have also been observed in cells from normally aged individuals and in age-associated pathologies including neurodegeneration (51, 52, 53).

Here, with the aim of exploring the potential contribution of NE dysfunction to other premature ageing pathologies, we observed loss of NE integrity in cells from Cockayne syndrome A patients. These cells displayed a reduction in the NE-associated LEMD2 protein in CS-A cells, resulting in decreased formation of lamin A/C-LEMD2 complexes at the NE. LEMD2 has a well-established role in the maintenance of NE morphology and cell survival (40, 54). Our findings are consistent with a previous study showing that LEMD2 depletion leads to NE defects without affecting the localization or expression of lamin A/C and lamin B1 (40). The authors speculated that although lamin proteins still localized to the NE, the lack of LEMD2 connected to the lamina network results in NE fragility. Because the LEMD2 interaction with lamin A/C is reduced in CS-A cells, this could contribute to destabilization of the lamina, causing NE instability. The NE phenotypes observed in CS-A cells also resemble those observed in Marbach–Rustad progeria syndrome (MRPS). MRPS is a recently characterized premature ageing disorder caused by a de novo mutation (c.1436C>T, p.S479F) in the C-terminal domain of LEMD2 (55). The mutation causes "patchy" localization of LEMD2 within the NE without affecting the total LEMD2 protein expression. Both MRPS and CS-A patient fibroblasts exhibit reduced nuclear circularity and increased nuclear blebbing phenotypes, further supporting the notion that reduced LEMD2 at the NE may underlie the pathogenic NE phenotype in CS-A cells.

Mutations in *LEMD2* have also been linked to cardiomyopathy in humans (56). For instance, patients carrying a homozygous mutation (c.38T>G, p.L13R) in the *LEMD2* gene develop ventricular arrhythmia and fibrosis (57). It is noteworthy to mention that the L13R mutation causes a significant reduction in LEMD2 expression in cardiomyocytes and causes NE defects including nuclear membrane invaginations and decreased nuclear circularity. MRPS patients with a mutation in *LEMD2* also display cardiovascular defects with septal hypertrophy and right bundle branch block. However, CSA patients do not display any form of cardiomyopathy. A study in mice by Ross et al (58) showed that reducing Lem2 levels in adult mice cardiomyocytes by ~45% did not lead to NE defects or cardiac dysfunction, suggesting a redundant function of LEMD2 at the NE in the mouse heart. As our data showed a reduction in LEMD2 (~20%) at the NE in CS-A cells compared with WT, one could hypothesize that the remaining LEMD2 at the NE is sufficient to maintain NE function in cardiomyocytes of CSA patients.

Recent work suggested the LEMD2-specific interactome included CUL4A, DDB1, and CSN, which together form an E3 ubiquitination ligase complex with CSA (41). Here, we showed by immunoprecipitation that LEMD2 also interacts with CSA. This could suggest that the recruitment and stabilization of LEMD2 to the NE is mediated by an interaction with CSA, although the mechanism remains unclear and would require further experimentation. Our FRAP

---

**Figure 3. Decreased NE localization of LEMD2 is a contributing factor to NE defects in CS-A cells.**
**(A)** Representative confocal images of DAPI and PLA signal in the indicated cells using either anti-lamin A/C or anti-LEMD2 antibodies alone (negative controls) or in combination; scale bar: 50 $\mu$m. **(B)** Quantification of the number of PLA foci per nuclei. *P*-value was calculated using a one-way ANOVA test followed by Tukey's post hoc test (****$P < 0.0001$), n = 3 with >100 cells per experiment. **(C)** Representative immunofluorescence staining of DAPI and lamin B1 in WT(HA-CSA) and CS-A cells transfected with GFP and LEMD2-GFP-containing constructs. **(D, E)** Quantification of the nuclear form factor and (E) percentage of nuclear blebs in GFP expressing cells, n = 3 with >100 cells per experiment. *P*-value was calculated using a one-way ANOVA test followed by Tukey's post hoc test (*$P < 0.05$, ***$P < 0.001$, ****$P < 0.0001$). All experiments in this figure were n = 3 independent experiments.

**A**

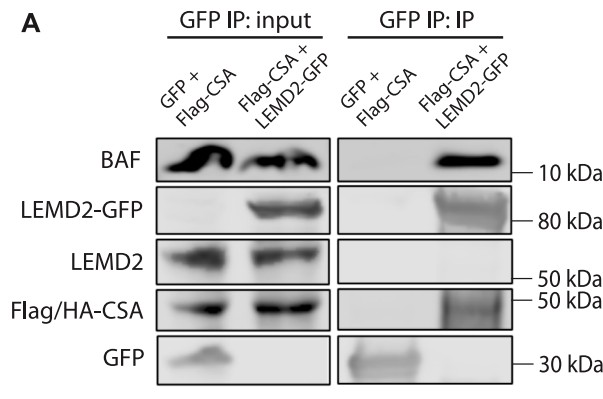

**B**

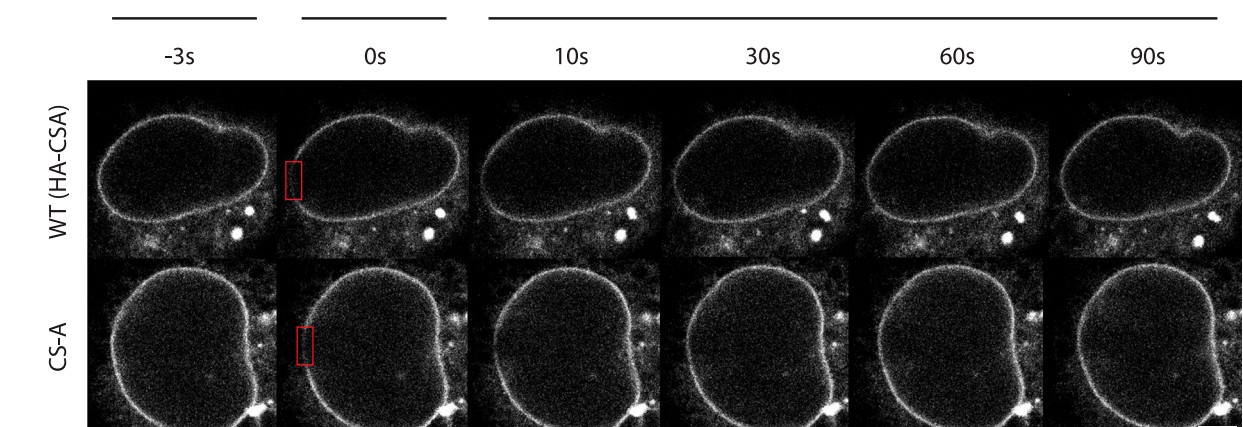

**C**

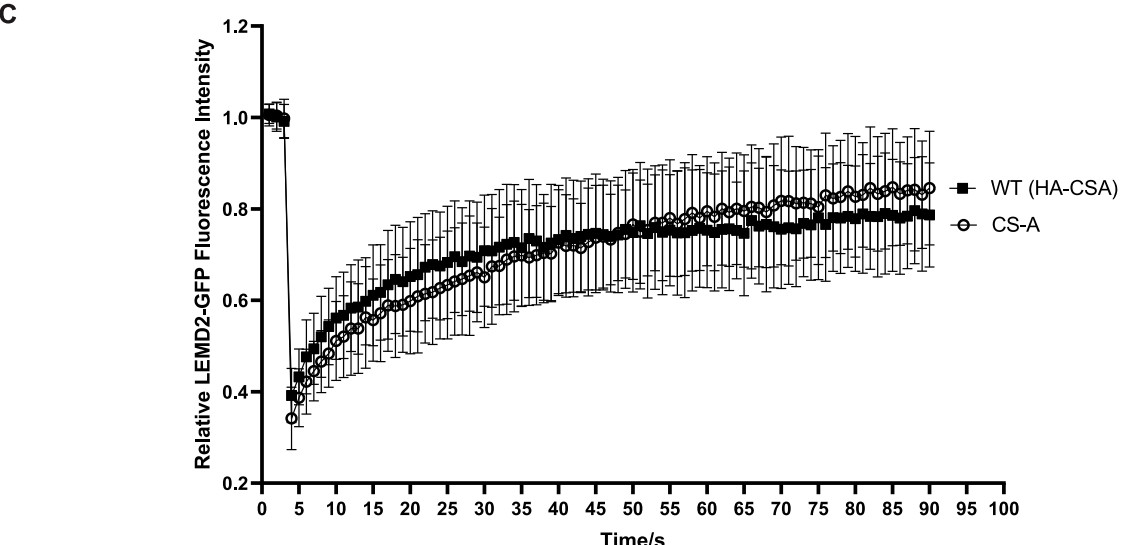

**D**

| Experiment | N | Mean Percentage of Immobile Fraction | | Mean Half-time of Recovery | |
|---|---|---|---|---|---|
| | | % ± SEM | Significance | Seconds ± SEM | Significance |
| WT (HA-CSA) + LEMD2-GFP | 3 | 77.22 ± 5.30 | No (p>0.05) | 10.72 ± 2.90 | No (p>0.05) |
| CS-A + LEMD2-GFP | 3 | 86.38 ± 4.38 | | 21.38 ± 2.95 | |

experiments showed that the presence or absence of CSA did not influence LEMD2 mobility. The potential caveat here is that the GFP-LEMD2 overexpression, required to carry out this experiment, may have been enough to rescue a normal NE composition in CS-A cells. An alternative method to bypass this would be to engineer endogenous tagging of LEMD2 with GFP, to allow the study of LEMD2 protein motility without overexpression.

In our search for new molecular mechanisms contributing to NE defects in CS-A cells, and because previous literature suggested that CS was associated with broad transcriptional changes, we analyzed publicly available RNA-seq data from WT(HA-CSA) and CS-A cells obtained in the absence of damage caused by UV irradiation (42). We found CS-A cells displayed a specific transcriptional dysregulation of cytoskeletal proteins. Our data confirmed an increase in actin stress fibers in CS-A cells. The contribution of actin cytoskeletal forces to spontaneous NE ruptures was established by a study conducted by Hatch and Hetzer (46). The authors showed that for cells growing on a flat and rigid 2D substrate (such as the ones used in standard culture conditions), contractile actin compresses the nucleus, leading to chromatin herniations and NE ruptures. Similarly, our findings show that disruption of actin polymerization was enough to improve the NE abnormalities in CS-A cells. On the other hand, chemically induced stabilization of the actin network in WT(HA-CSA) cells induced NE defects similar to that of the untreated CS-A cells. These data have reinforced the role of actin stress bundles in inducing spontaneous NE ruptures and have also highlighted how actin stress fibers in CS-A cells, occurring through transcriptional deregulation, contribute to the appearance of NE deformation and ruptures.

Another way to release mechanical stress on the nucleus is to disrupt the connection between actin and the NE by interfering with the integrity of the LINC complex. As such, depletion of SUN1—a major component for LINC complex assembly—has been reported to correct several pathological phenotypes in HGPS and lamin A/C-deficient cells including nuclear shape, NE blebbing, heterochromatin loss, chromatin disorganization, and cellular senescence (59). In vivo data also showed that removal of SUN1 in HGPS and lamin A/C-deficient mouse models improved longevity and multiple pathological phenotypes including body weight deficit, lordokyphosis, trabecular, and bone densities (59). Similarly, SUN1 depletion led to the improvement of NE phenotypes in CS-A cells, reinforcing the idea that forces exerted by actin on the nucleus of CS-A cells contribute to the NE abnormalities and ruptures in these cells. These data also suggest that disrupting a component of the LINC complex could be an effective strategy to restore cellular homeostasis in multiple syndromes associated with NE dysfunction, and it is an approach that is currently being investigated by other teams (60, 61).

As a result of the NE ruptures in CS-A cells, we observed the activation of the innate immune cGAS/STING pathway which was reduced upon SUN1 depletion in CS-A cells. Activation of cGAS/STING can induce the transcription factor NF-κB which in turn up-regulates the release of proinflammatory chemokines, cytokines, and growth factors (48, 62). These immune modulators act in an autocrine and paracrine manner to induce propagation and amplification of senescence in distant cells, which is a characteristic of senescence-associated secretory phenotype (63). In many diseases, senescence-associated secretory phenotype is the main contributor to chronic inflammation and progression of fibrosis (64, 65). The only in vivo study suggesting inflammatory phenotype in CS model was performed using *CSA* KO (CX) mice (66). They observed increased senescence of brain endothelial cells and an up-regulation of proinflammatory markers in the brains of the CX mice including ICAM-1, TNF-a, and p-p65. Establishing the validity of the neuroinflammation phenotype in CS and the potential link with NE defects observed in CS-A cells using iPSC-derived or trans-differentiated neuronal CSA patient cell lines would be an interesting area of investigation.

Overall, our study has identified a new, non-canonical function of CSA in regulating NE integrity independent of its established role in the TC-NER pathway. This further reinforces the role played by NE dysfunction in various age-related conditions, outside of the known laminopathies. In CS-A cells, NE fragility and rupturing may contribute to the accumulation of DNA damage over time and the activation of innate immune pathways, potentially contributing to inflammation in tissues such as the brain. Further understanding of the mechanisms behind the NE defects in CSA patient-derived cells may help explain some of the clinical phenotypes observed in CS patients and open new therapeutic avenues.

# Materials and Methods

### Cell lines and cell culture

The CSA patient-derived fibroblast cells (CS3BE, termed CS-A in this study), CSB patient-derived fibroblast cells (CS1AN, termed CS-B in this work), and their respective isogenic cell lines (complemented with HA-CSA—WT(HA-CSA) or HA-GFP-CSB—WT(HA-GFP-CSB), all immortalized with hTERT, were a kind gift from Dr. Sebastian Iben. WT fibroblasts were derived from a healthy individual (AG10803; Coriell repositories) and immortalized with SV40LT and TERT. Stable Cas9 expression was engineered through transduction with purified lentiviral particles, containing a vector encoding the *S. pyogenes* Cas9 nuclease under the control of an hCMV promotor and according to the manufacturer's protocol (#VCAS10124; Horizon Discovery). Cells were grown in a complete medium consisting of

**Figure 4. The absence of CSA does not affect LEMD2 mobility at the NE.**
**(A)** Immunoprecipitation of LEMD2-GFP in WT(HA-CSA) cells overexpressing GFP+Flag-CSA (control) and Flag-CSA+LEMD2-GFP. **(B)** Representative time-lapse confocal images showing pre-bleach, bleach, and post-bleach images of the WT(HA-CSA) and CS-A nuclei with LEMD2-GFP overexpression. The red box indicates the photobleached region at the nuclear periphery. **(C)** Graph showing the FRAP kinetics of LEMD2-GFP expressed in WT(HA-CSA) (n = 25 cells) and CS-A (n = 25 cells) cells. Error bars represent SD. Scale bar: 5 μm. **(D)** Summary table showing the mean percentage of the immobile fraction and the mean half-time of recovery of LEMD2-GFP FRAP experiment. Data were compared using a two-tailed unpaired *t* test. All experiments in this figure were n = 3 independent experiments. Source data are available for this figure.

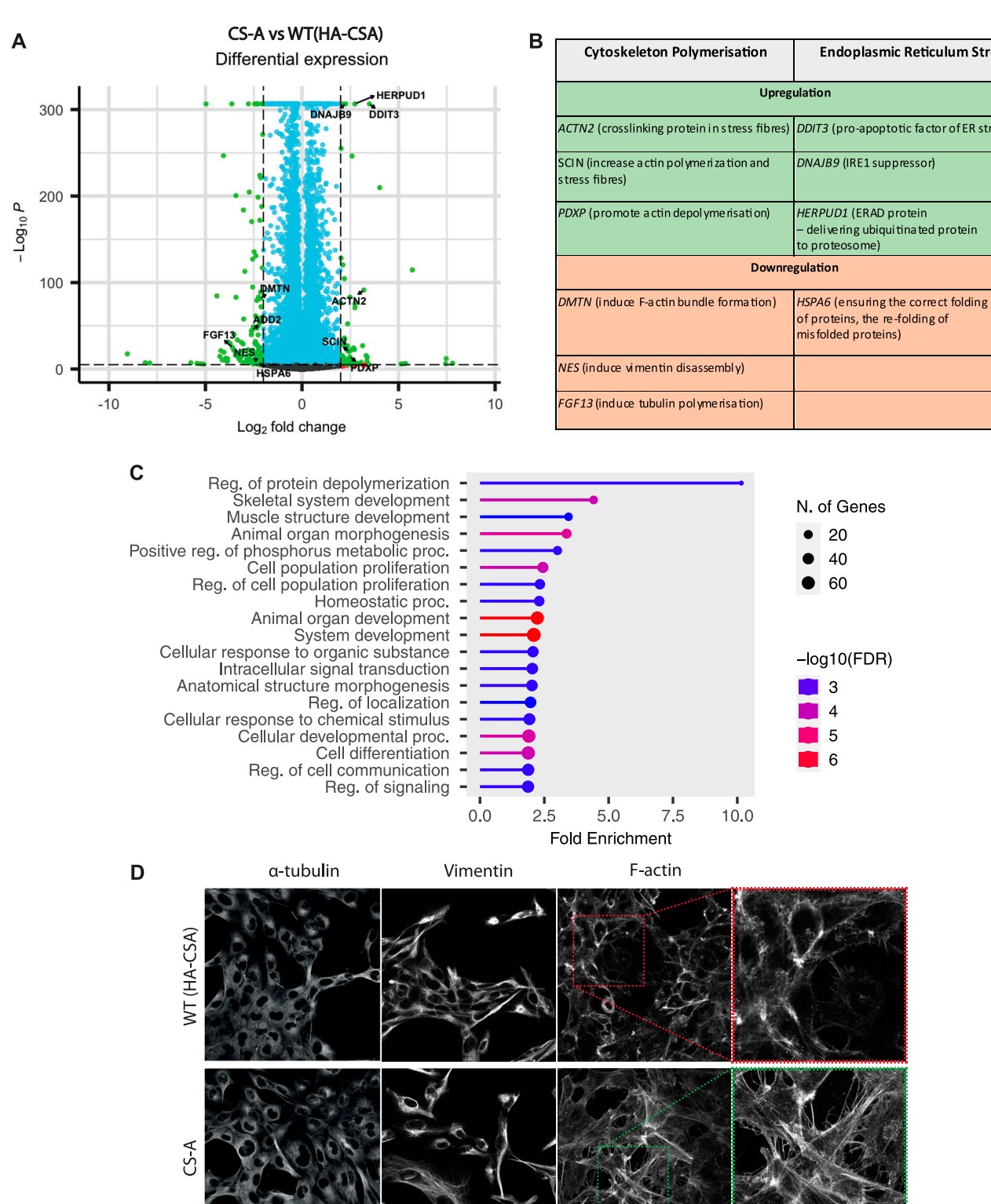

**Figure 5. Loss of CSA results in transcriptional deregulation of cytoskeleton polymerization-regulating genes.**
**(A)** Volcano plot showing differentially expressed genes in CS-A and WT(HA-CSA) cells. **(B)** Table showing significantly up-regulated and down-regulated genes involved in cytoskeleton polymerization and ER stress. **(C)** GO enrichment analysis of differentially expressed genes comparing CS-A and WT(HA-CSA) showing top 20 enriched GO terms in biological process. **(D)** Representative immunofluorescence staining showing the appearance of F-actin stress fibers in CS-A cells compared with WT(HA-CSA), but no noticeable difference in α-tubulin, and vimentin networks; scale bar: 50 μm. All experiments in this figure were n = 3 independent experiments.

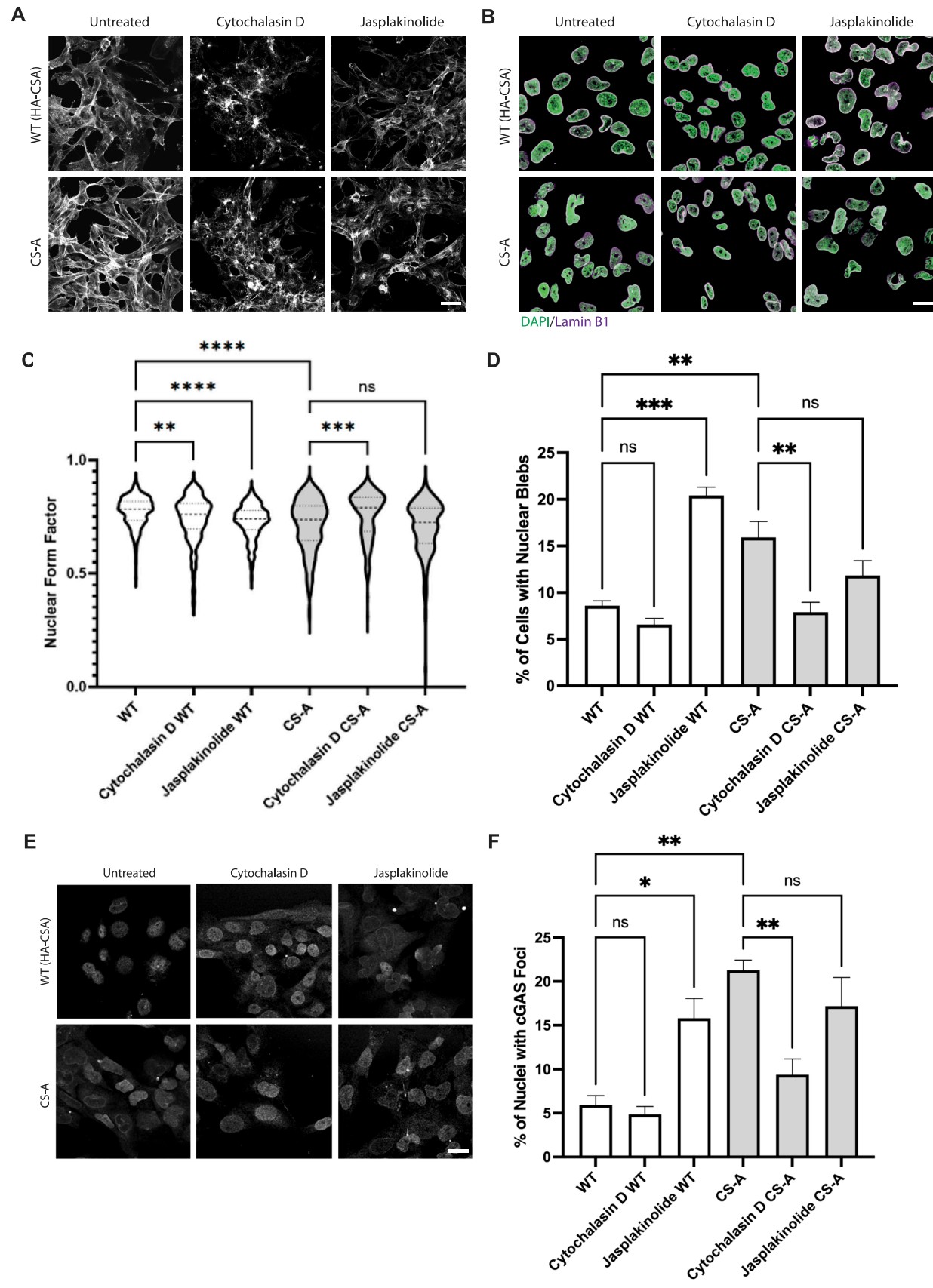

Dulbecco's Modified Essential Medium with 10% (vol/vol) FBS (Gibco) and 1% (vol/vol) penicillin/streptomycin (10,000 U/ml) (Thermo Fisher Scientific). The complemented WT(HA-CSA) and WT (HA-GFP-CSB) cells were grown in complete media with 50 µg/ml of Geneticin (Life Technologies) as a selection media to maintain HA-CSA or HA-GFP-CSB expression. AG10803 cells were grown in complete media with 5 µg/ml blasticidin (Life Technologies) as a selection media to maintain Cas9 expression. *CSA* and *CSB* KO HAP1 cell lines were purchased from Horizon Discovery and cultured in Iscove's Modified Essential Medium media, containing 10% FBS and 1% (vol/vol) penicillin/streptomycin (10,000 U/ml).

### Drug treatments

For drug treatments, cells were seeded onto 12-well plate at 50% confluency. Drug concentration and treatment time used throughout this study are outlined in Table 1. At the start of each drug treatment, culture media was replaced with fresh media containing the working concentration of the indicated drugs (Table 1).

### Plasmid transfection and constructs

Cells were first seeded onto the tissue culture plate at 70% confluency. The next day, transfection of plasmid constructs was performed using Trans-IT 2020 reagent (#MIR 5400; Mirus Bio) following the manufacturer's protocols. 24 h after transfection, experiments were performed. pEGFP-C1 plasmid construct was purchased from Clontech. Flag-CSA construct subcloned in pcDNA3.1 vector was a kind gift Dr. Sebastian Iben. LEMD2-GFP was a kind gift from Dr. Kyle Roux (33).

### siRNA knockdown

The siRNA oligonucleotides were purchased from Sigma-Aldrich with the following sequences: siSUN1: 5'-CCAUCCUGAGUAUACCU-GUCUGUAUDTDT-3', siLEMD2: 5'-UUGCGGUAGACAUCCCGGGDTDT-3'. MISSION siRNA Universal Negative Control #1 (#SIC001; Sigma-Aldrich) was used as the control siRNA. ON-TARGETplus siRNA oligonucleotides in SMARTpool format targeting SUN2 (L-009959-01-0005) and Nesprin1 (L-014039-00-0005) were purchased from Horizon. Cells were first seeded onto a tissue culture plate at 50–60% confluency. The next day, the transfection of siRNA was performed using Lipofectamine RNAiMAX (#13778075; Thermo Fisher Scientific) following the manufacturer's instructions. 30 pmol of siRNA were transfected into cells in 12-well plates for 48 h.

### Single guide RNA transfection

Single guide RNA (sgRNA) targeting *ERCC8* was purchased from Synthego with the following sequence: 5'-UUUAUUAUCAGCAU-GUUAUC-3'. *ERCC8* gene was knocked out in Cas9-expressing WT AG10803 fibroblast cells by reverse transfecting 120,000 cells with 30 pmols of *ERCC8*-targeting sgRNA in 12-well plate format using DharmaFECT-1 reagent (Horizon) following manufacturer's instructions. Cells were harvested 72 h later.

### Western blotting

Protein extraction from monolayered culture cells was performed by scraping cells in SDS lysis buffer (4% SDS, 20% glycerol, 120 mM Tris–HCl, pH = 6.8), boiling for 5 min at 95°C, and passing through a 25-gauge needle 10 times. For samples requiring pre-extraction of soluble proteins, cells were first incubated with cold cytoskeletal buffer (CSK) (100 mM NaCl, 300 mM sucrose, 1 mM EGTA, 1 mM MgCl$_2$, 1 mM DTT, 10 mM PIPES/KOH, 6.8 pH) for 5 min on ice. Protein concentration was measured using the nanodrop spectrophotometer (Thermo Fisher Scientific) at 280 nm. 30 µg of each protein sample was then heat-denatured for 5 min at 95°C after adding Protein Sample Loading Buffer (#928-40004; LI-COR) with 100 mM DTT. Denatured protein samples and PageRuler Prestained Protein ladder (#26616; Thermo Fisher Scientific) and PageRuler Plus Prestained Protein ladder (#26619; Thermo Fisher Scientific) were size-separated using precast NuPAGE 4–12%, Bis-Tris, 1.0-1.5 mm, Mini Protein Gels (#NP0322BOX; Thermo Fisher Scientific) in 1X NuPAGE MES SDS Running Buffer (#NP0002; Thermo Fisher Scientific) at 180 V. Size-separated proteins were then transferred onto a nitrocellulose or PVDF membrane in 1X NuPAGE Transfer Buffer (#NP0006; Thermo Fisher Scientific) at 250 mA. SDS–PAGE and protein transfer were performed using the Mini Gel Tank (#A25977; Thermo Fisher Scientific) and Mini Blot Module (#B1000; Thermo Fisher Scientific) system, respectively. The nitrocellulose membrane was blocked with 5% (wt/vol) non-fat milk dissolved in TBS (50 mM Tris–HCl, 150 mM NaCl) with 0.1% (vol/vol) Tween-20 (Sigma-Aldrich) (0.1% TBS-T) for 1 h at room temperature with gentle agitation. Primary antibody incubation was performed for either 1 h at room temperature or overnight at 4°C. Secondary antibody incubation was performed for 1 h at room temperature. Details of primary and secondary antibodies used are outlined in Table 2. Protein detections were carried out using the Odyssey CLx Imager (LI-COR) system. For densitometric analysis, band intensities of the protein of interest were normalized to the band intensity of housekeeping protein and

---

**Figure 6. Modulation of actin polymerization affects the NE phenotypes in CS-A cells.**
**(A)** Immunofluorescence staining of F-actin in WT(HA-CSA) and CS-A cells treated with 0.25 µg/ml cytochalasin D and 25 nM Jasplakinolide for 24 h; scale bar: 50 µm. **(B)** Representative immunofluorescence staining of DAPI and lamin B1 in WT(HA-CSA) and CS-A cells treated with the actin polymerization inhibitor (0.25 µg/ml cytochalasin D) for 6 h or the actin stabilizer (25 nM Jasplakinolide) for 16 h; scale bar: 25 µm. **(C, D)** Quantification of the nuclear form factor and (D) percentage of nuclear blebs in cytochalasin D and Jasplakinolide-treated cells, n = 3 with >100 cells per experiment. *P*-value was calculated using a one-way ANOVA test followed by Tukey's post hoc test (ns *P* > 0.05, **P* < 0.01, ***P* < 0.001, ****P* < 0.0001). **(E)** Representative immunofluorescence staining of cGAS in WT(HA-CSA) and CS-A cells treated with the actin polymerization inhibitor (0.25 µg/ml cytochalasin D) for 6 h or the actin stabilizer (25 nM Jasplakinolide) for 16 h. **(F)** Quantification of the percentage of nuclei with cGAS foci in cytochalasin D and Jasplakinolide-treated cells, n = 3 with >100 cells per experiment. *P*-value was calculated using a one-way ANOVA test followed by Tukey's post hoc test (ns *P* > 0.05, **P* < 0.01). All experiments in this figure were n = 3 independent experiments.

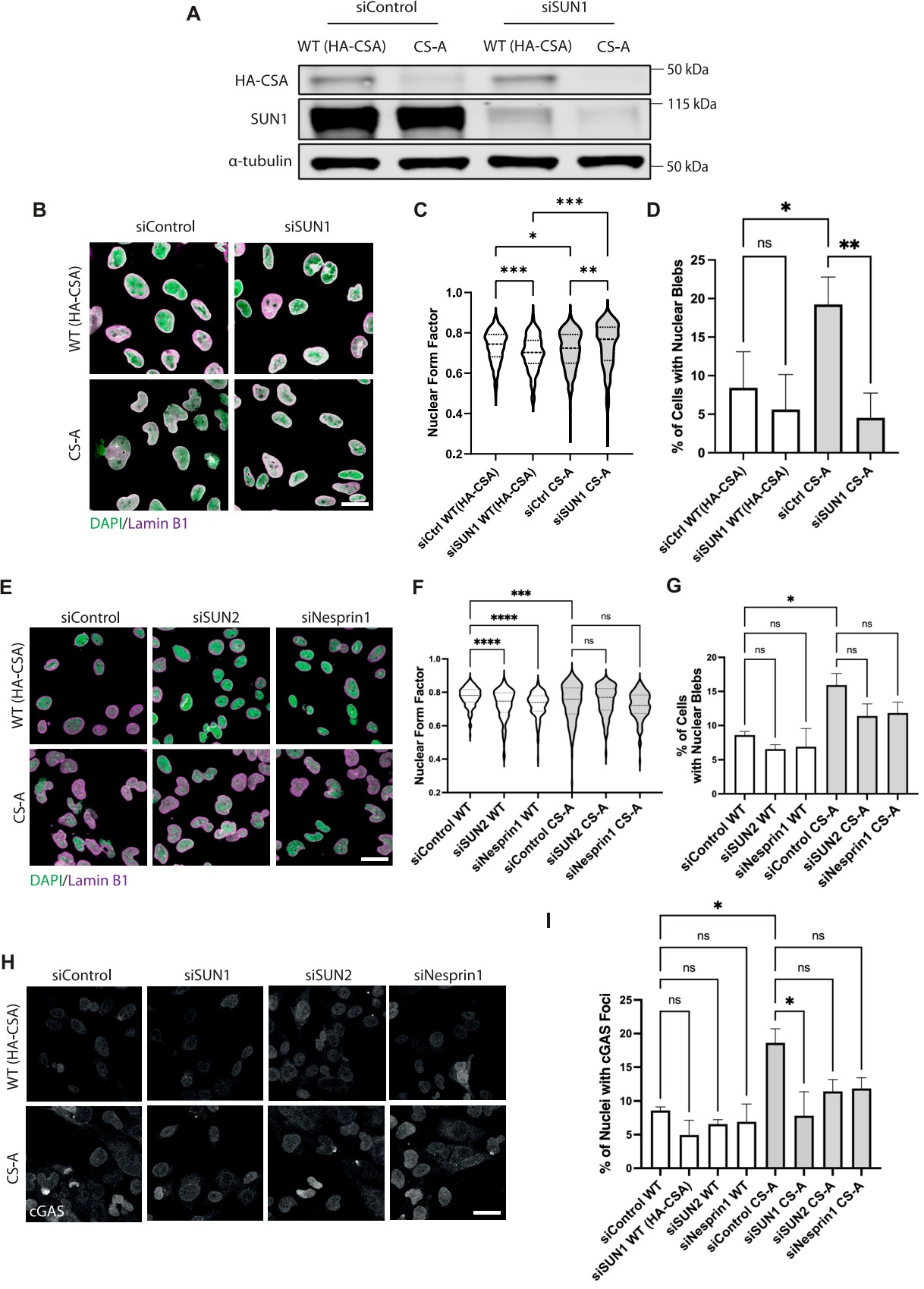

followed by normalizing all tested samples to the WT or control samples.

## Immunoprecipitation (IP)

Cells were lysed in IP lysis buffer (0.5% NP-40, 150 mM NaCl, 0.5 mM EDTA, 10 mM Tris–HCl, pH = 7.4) with freshly prepared 0.5 mM PMSF and cOmplete Mini EDTA-free Protease Inhibitor Cocktail (#04693159001; Roche) on ice for 30 min with gentle vortexing every 10 min. Lysates were subjected to centrifugation at 15,000 *g* for 15 min at 4°C and the supernatant was collected. The remaining pellet was then resuspended in RIPA buffer freshly supplemented with 0.5 mM PMSF and cOmplete Mini EDTA-free Protease Inhibitor Cocktail and subjected to sonication two times at 10 kHz for 10 s. The resuspended pellet solution was then centrifuged at 15,000 *g* for 15 min at 4°C and the supernatant was combined with the supernatant obtained from the first lysis. Input samples were collected from the combined lysates. For the pulldowns, ChemTek GFP-Trap Magnetic Agarose (#gtma; Proteintech) or ANTI-FLAG M2 Affinity Gel (#A2220; Sigma-Aldrich) were incubated with the protein lysates for 4 h under rotation at 4°C. Protein-bound beads were then washed four times with wash buffer. Protein elution from the protein-bound beads was performed by adding Protein Sample Loading Buffer (#928-40004; LI-COR) with 10 mM DTT followed by boiling for 5 min at 95°C.

## PLA

Cells were first seeded onto 12 mm coverslips and fixed in 4% (wt/vol) paraformaldehyde dissolved in PBS for 10 min at room temperature. Permeabilization was performed by incubating cells with 0.2% (wt/vol) Triton X-100 dissolved in PBS for 15 min at room temperature. The following steps including blocking, primary antibody incubation, DuoLink Probe incubation, ligation, and amplification were performed using the Duolink PLA assay kit (#DUO92008; Sigma-Aldrich) following the manufacturer's instructions. Primary antibodies for incubation were mouse anti-lamin A/C (#sc-376248, 1:1,000; Santa Cruz) and rabbit anti-LEMD2 (#HPA017340, 1:250; Sigma-Aldrich). For the Duolink In Situ PLA probe, anti-mouse MINUS and anti-rabbit PLUS were used. For amplification, Duolink Amplification red was used. Coverslips were then mounted using Duolink Mounting Media containing DAPI. Acquisition of microscopy images was performed with the Stellaris 5 confocal laser scanning microscope using the 40X oil immersion objective lens (Leica HC PLA APO, 1.3 NA).

## Immunofluorescence assay

For samples requiring pre-extraction of soluble proteins before immunofluorescence assays, cells were incubated with cytoskeletal buffer (CSK) (100 mM NaCl, 300 mM sucrose, 1 mM EGTA, 1 mM MgCl$_2$, 1 mM DTT, 10 mM PIPES/KOH, 6.8 pH) for 5 min on ice. Cells were fixed in 4% (wt/vol) paraformaldehyde dissolved in PBS for 10 min at room temperature. Permeabilization was performed with 0.2% (vol/vol) Triton X-100 in PBS for 15 min at room temperature. Blocking was performed with 2% (wt/vol) BSA dissolved in PBS for 30 min at room temperature. Coverslips were then incubated with primary antibodies and secondary antibodies, both for 1 h at room temperature. Details of primary antibodies and secondary antibodies are outlined in Table 2. Coverslips were mounted onto glass slides using Prolong Gold Antifade Reagent (#9071; Cell Signalling). Acquisition of microscopy images was performed with the Stellaris 5 confocal laser scanning microscope using the 40X oil immersion objective lens (Leica HC PLA APO, 1.3 NA).

## Automated CellProfiler workflow

CellProfiler V4.1.3 software was used to analyze images obtained from immunofluorescence assays to NE phenotypes, protein localization, and protein fluorescent signal intensity. All Cell Profiler pipelines used in this work can be made available upon request.

To assess nuclear morphology, the nuclear form factor was calculated for each identified nuclei to determine its roundness. The nuclear form factor is calculated by $4\pi A/P^2$, where A is the area of the nucleus, and P is the perimeter of the nucleus. A perfectly round nucleus has a form factor of 1 and a more deformed nucleus will have a form factor lower than 1. DAPI staining was used to identify individual nuclei using the "Identify Primary Objects" setting, and border objects were excluded. After the identification of nuclei, the nuclear form factor was calculated using the "Measure Object Size and Shape" setting. To measure the LEMD2 intensity at the nucleus, the measurement was performed by measuring the signal intensity of the LEMD2 channel in the identified nucleus, defined by the DAPI mask. To identify the cGAS foci at the nucleus, the signal intensity of cGAS foci was enhanced relative to the rest of the image using the "Enhance Or Suppress Features" setting. cGAS foci were then identified using the "Identify Primary Objects" setting.

The pipeline used to identify nuclear bleb was a modified version of the pipeline described before (38). DAPI staining and lamin B1 staining were used to identify individual nuclei using the "Identify

**Figure 7. Disruption of SUN1 improves NE phenotypes in CS-A cells.**
**(A)** Immunoblot showing the efficiency of SUN1 knockdown by siRNA in WT(HA-CSA) and CS-A cells. **(B)** Representative immunofluorescence staining of DAPI and lamin B1 in WT(HA-CSA) and CS-A cells transfected with SUN1-targeting siRNA (siSUN1); scale bar: 25 $\mu$m. **(C, D)** Quantification of the nuclear form factor and (D) percentage of nuclear blebs, n = 3 with >100 cells per experiment. siCtrl: scramble siRNA, siSUN1: siRNA targeting SUN1. *P*-value was calculated using a one-way ANOVA test followed by Tukey's post hoc test (ns *P* > 0.05, \**P* < 0.05, \*\**P* < 0.01, \*\*\**P* < 0.001). **(E)** Representative immunofluorescence staining of DAPI and lamin B1 in WT(HA-CSA) and CS-A cells transfected with siRNA targeting SUN2 (siSUN2) and Nesprin1 (siNesprin1); scale bar: 25 $\mu$m. **(F, G)** Quantification of the nuclear form factor and (G) percentage of nuclear blebs in siSUN2 and siNesprin1 cells, n = 3 with >100 cells per experiment. *P*-value was calculated using a one-way ANOVA test followed by Tukey's post hoc test (ns *P* > 0.05, \**P* < 0.05, \*\*\**P* < 0.001, \*\*\*\**P* < 0.0001). **(H)** Representative immunofluorescence staining of cGAS in siSUN1, siSUN2, and siNesprin1 treated WT(HA-CSA) and CS-A cells; scale bar: 25 $\mu$m. **(I)** Quantification of the percentage of nuclei with cGAS foci in siSUN1, siSUN2, and siNesprin1 treated WT(HA-CSA) and CS-A cells, n = 3 with >100 cells per experiment. *P*-value was calculated using a one-way ANOVA test followed by Tukey's post hoc test (ns *P* > 0.05, \**P* < 0.05).
Source data are available for this figure.

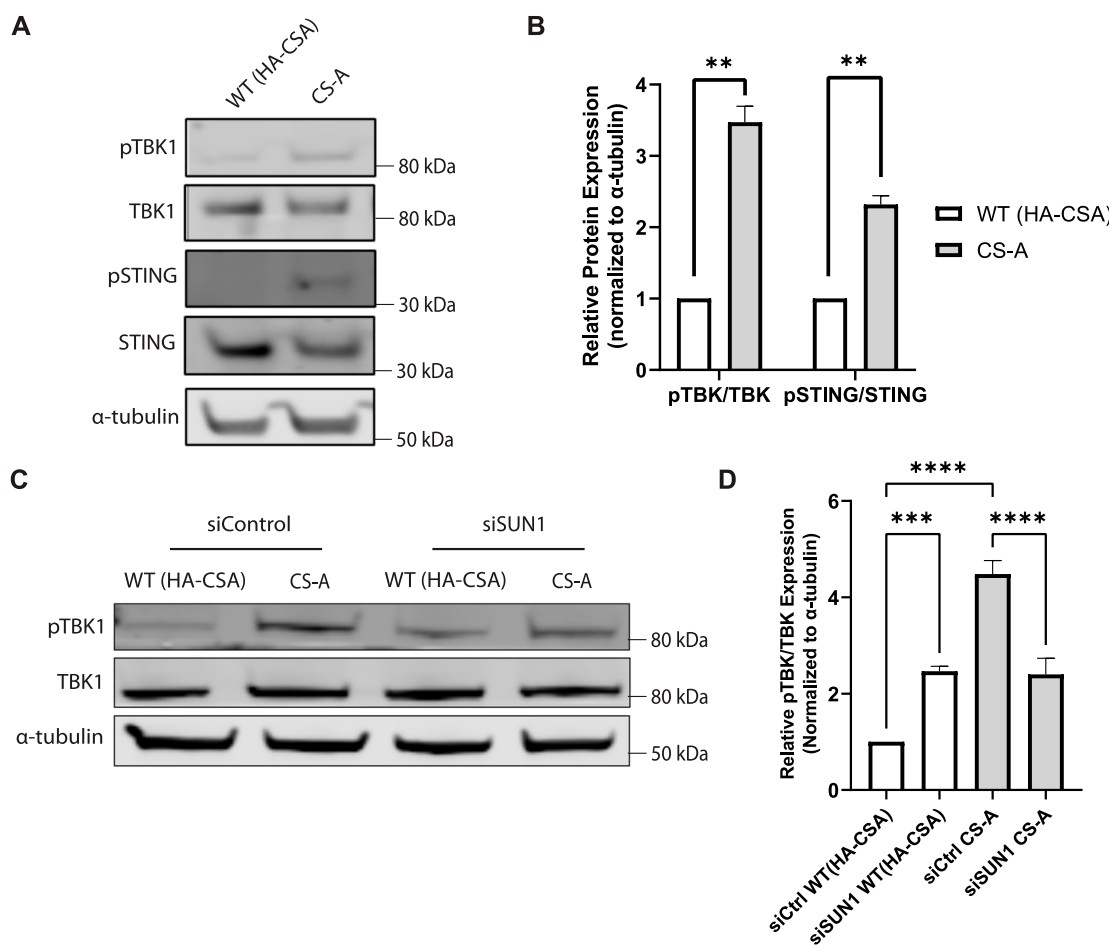

**Figure 8. SUN1 depletion reduces the activation of the cGAS/STING pathway in CS-A cells.**
**(A)** Representative western blot showing phosphorylation levels of TBK1 and STING in WT(HA-CSA) and CS-A cells. **(B)** Quantification of the relative phosphorylation level of cGAS-STING pathway proteins. *P*-value was calculated using a paired *t* test (**P < 0.01). **(C)** Representative western blot showing phosphorylation levels of TBK1 in WT(HA-CSA) and CS-A cells upon SUN1 depletion. **(D)** Quantification of the relative phosphorylation level of TBK1 in SUN1-depleted cells *P*-value was calculated using a one-way ANOVA test followed by Tukey's post hoc test (***P < 0.001, ****P < 0.0001). All experiments in this figure were n = 3 independent experiments. Source data are available for this figure.

Primary Objects" setting, and border objects were excluded. Blebs were identified by subtracting the pixel areas of DAPI and lamin B1 staining using the "Identify Tertiary Objects" setting. Identified blebs were then filtered by minimum pixel area to remove small identified false blebs and edge pixels of the nuclei.

To quantify nuclear morphology and nuclear blebs in GFP-expressing or LEMD2-GFP–expressing cells, individual nuclei were first identified using DAPI staining with the "identify Primary Objects" setting. Then, the GFP or LEMD2-GFP–positive nuclei were selected in the GFP channel using the "MaskObjects" setting. Nuclear morphology and nuclear blebs of GFP or LEMD2-GFP–positive nuclei were quantified as described above.

To quantify PLA foci, individual nuclei were first identified using "Identify Primary Objects" setting, and border objects were excluded. Signal intensity of PLA foci was enhanced relative to the rest of the image using the "Enhance Or Suppress Features" setting before the PLA foci were identified using the "Identify Primary Objects" setting.

## RNA sequencing and differentially expressed gene analysis

The read count matrix of WT(HA-CSA) and CS-A was downloaded from the Gene Expression Omnibus (GEO) website (GSE87540) (42). The datasets were obtained from WT(HA-CSA) and CS-A cells grown in DMEM/HamF10 media with 10% (vol/vol) fetal calf serum and in 40 μg/ml gentamycin, without UV irradiation.

The Deseq2 R package (v.1.24.0) was used to analyze the publicly available RNA-seq data unless otherwise stated. The "Median of ratios" method was used to normalize the raw count data using the "estamateSizeFactors" function. To compare the gene expression level between CS-A and WT, $\log_2$ fold change and *P*-values were calculated using the "nbinomWaldTest" function. The *P*-values obtained from the Wald tests were adjusted for multiple testing using the Benjamini-Hochberg method. The list of significantly up-regulated and down-regulated genes was determined by subsetting differentially expressed (DE) genes with adjusted *P*-values less than 0.001 and

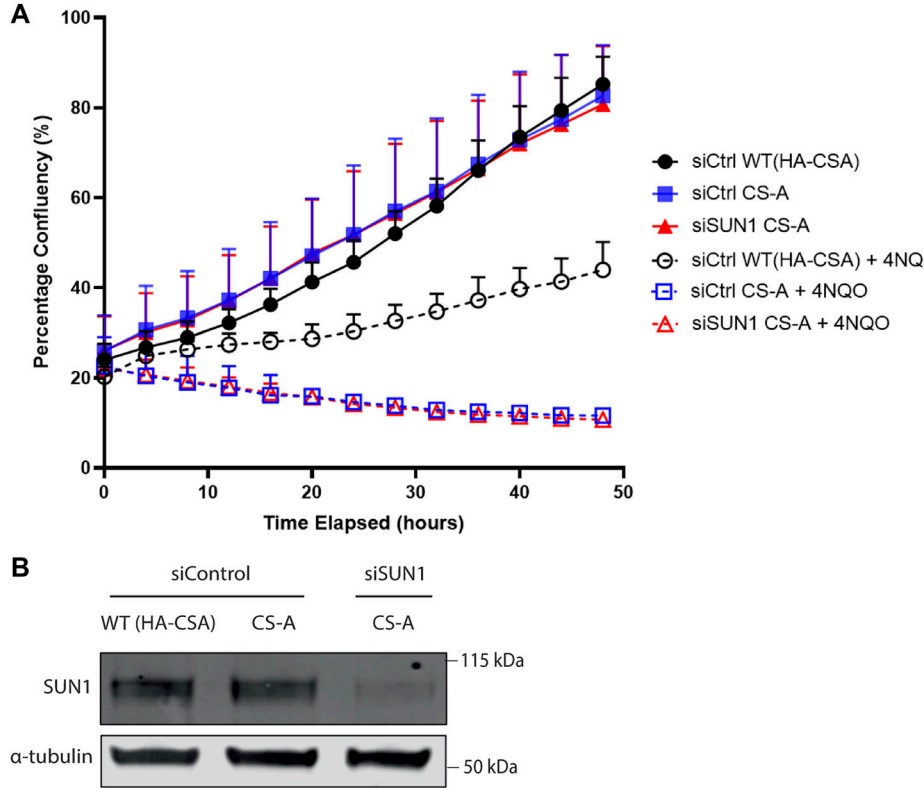

**Figure 9. SUN1 depletion does not rescue the UV sensitivity phenotype of CS-A cells.**
**(A)** WT(HA-CSA) and CS-A cells transfected with control and SUN1 siRNA were treated for 1-h with 4NQO, then imaged every 4 h for 48 h using an Incucyte system. Cell confluency was assessed with the Incucyte software. Error bars represent SD. **(A, B)** Immunoblot showing the depletion of SUN1 in CS-A cells as presented in (A). Source data are available for this figure.

**Table 1. Details of chemical used for drug treatment experiments.**

| Drug | Working concentration | Incubation time/hour | Manufacturer | Catalogue number |
|---|---|---|---|---|
| Cytochalasin D | 0.25 μg/ml | 18 h | Cambridge Bioscience | 11330 |
| Jasplakinolide | 25 nM | 6 h | Abcam | ab141409 |

absolute value of log$_2$ fold change greater than 2. Gene ontology (GO) analysis and STRING protein-protein interaction diagram of the DE genes were performed using the STRING website and ShinyGO 8.0 (67). The volcano plot was generated using the "EnhancedVolcano" function from the EnhancedVolcano R Package to show significant DE genes between CS-A and WT. The adjusted P-values and log$_2$ fold change cut-off were the same as per subsetting significantly up-regulated and down-regulated genes.

## FRAP

WT(HA-CSA) and CS-A cells transiently expressing LEMD2-GFP fusion protein were imaged using 60X water-immersion objective lens (1.2NA; Zeiss) Zeiss 880 Airyscan Confocal microscope. Cells were seeded onto a μ-slide four well-chambered coverslip (#80426; ibidi) in CO$_2$ independent medium (#11580536; Gibco) containing 10% (vol/vol) FBS and imaged at 37°C using the microscope cage incubator. The fluorescence intensity of the ROI was measured over 93 s at 1 s interval (93 images in total) using 2% laser power from 488 nm light. Photobleaching was started

after three scans and recovery was followed by 90 scans. For photobleaching, 2.52 × 6.30 μm region of LEMD2-GFP fluorescence at the nuclear periphery in the mid–focal plane was photobleached by scanning 50 iterations using 100% light intensity from 488 nm light. The pinhole size was set at 1 AU for the confocal.

Analysis of FRAP data was performed using Microsoft Excel Version 16.43 and GraphPad Prism Version 9. Fluorescence intensity measurement of ROI was corrected by the fluorescence intensity of an unbleached area at the nuclear periphery to account for background bleaching. The data were fitted with a curve of the form $y = y_0 + (a - y_0)(1 - e^{-bx})$, where $(a,b)$ corresponds to the asymptotic values of relative LEMD2-GFP fluorescence intensity and the decay rate of growth, respectively. The percentage of immobile fraction ($IF\%$) was determined by $IF\% = a \cdot 100\%$ and half-time of recovery ($t_{1/2}$) was determined by $t_{1/2} = \frac{\ln(2)}{b}$.

## Live cell proliferation assay

WT(HA-CSA) and CS-A cells were seeded onto 24-well plates at ~35% confluency and transfected with either control siRNA or

**Table 2.  Details of primary and secondary antibodies used for immunofluorescence and Western blot experiments.**

| Primary antibodies | | | | |
|---|---|---|---|---|
| Antibody | Western blot dilution | Immunofluorescence dilution | Supplier | Catalogue number |
| Anti-LEMD2 | 1:200 | 1:200 | Sigma-Aldrich | HPA017340 |
| Anti-SUN1 | 1:1,000 | 1:100 | Abcam | ab124770 |
| Anti-α-tubulin | 1:500 | 1:500 | Sigma-Aldrich | T9026 |
| Anti-cGAS | 1:250 | 1:250 | Cell Signalling | D1D3G |
| Anti-TBK1 | 1:500 | | Cell Signalling | 3504 |
| Anti-pTBK1 | 1:500 | | Cell Signalling | 5483 |
| Anti-STING | 1:500 | | Cell Signalling | 13647 |
| Anti-pSTING | 1:500 | | Cell Signalling | 50907 |
| Anti-CSA | 1:500 | | Abcam | ab137033 |
| Anti-lamin A/C | 1:1,000 | | Santa Cruz | sc-7292 |
| Anti-histone H3 | 1:1,000 | | Cell signalling | 3638 |
| Anti-GFP | 1:1,000 | | Thermo Fisher Scientific | MA5-15256 |
| Anti-BAF | 1:250 | | ProSci | 4019 |
| Anti-CSB | 1:500 | | Abcam | ab96089 |
| Anti-DDB1 | 1:250 | | BD Biosciences | 612488 |
| Anti-GAPDH | 1:5,000 | | Invitrogen | MA5-15738 |
| Anti-lamin B1 | | 1:500 | Santa Cruz | sc-365214 |
| Anti-lamin A/C | | 1:1,000 | Santa Cruz | sc-376248 |
| Anti-emerin | | 1:400 | Cell Signalling | 30853 |
| Anti-vimentin | | 1:100 | Cell signalling | 5741 |
| CF488A-phalloidin | | 1:100 | Biotum | 00042-T |
| Anti-SUN1 | 1:1,000 | | Abcam | ab124770 |
| Anti-SUN2 | 1:1,000 | | Abcam | ab124916 |
| Anti-Nesprin1 | 1:500 | | Thermo Fisher Scientific | MA5-18077 |
| Secondary antibodies | | | | |
| Antibody | Western blot dilution | Immunofluorescence dilution | Supplier | Catalogue number |
| IRDye 800RD anti-mouse | 1:10,000 | | LI-COR | 925-32212 |
| IRDye 600 RD anti-rabbit | 1:10,000 | | LI-COR | 925-68073 |
| Alexa Fluor 647 anti-rabbit | | 1:1,000 | Life Technologies | A31573 |
| Alexa Flour 488 anti-mouse | | 1:1,000 | Life Technologies | A21141 |
| Alexa Fluor 568 anti-mouse | | 1:1,000 | Life Technologies | A21124 |
| Alexa Flour 488 anti-rabbit | | 1:1,000 | Life Technologies | A21206 |
| Alexa Fluor 647 anti-mouse | | 1:1,000 | Life Technologies | A21242 |
| Alexa Flour 488 anti-mouse | | 1:1,000 | Life Technologies | A21121 |
| Alexa Fluor 568 anti-rabbit | | 1:1,000 | Life Technologies | A10042 |
| Alexa Fluor 647 anti-mouse | | 1:1,000 | Life Technologies | A31571 |

SUN1-targeting siRNA. To induce bulky DNA adducts mimicking those caused by UV irradiation, cells were treated with 1.5 $\mu$M of 4-nitroquinoline 1-oxide (4NQO) for 1 h before imaging. Phase contrast images were acquired every 4 h over a period of 48 h using an Incucyte machine (Sartorius). The percentage of cell confluence was calculated using the attached Incucyte software (Sartorius).

## Statistical analysis

All statistical tests and graphs were generated using GraphPad Prism Version 9. Error bars in graphs are shown as mean ± SD. Post hoc tests were performed for experiments that required correction for multiple comparisons. Details of specific statistical tests are described in the figure legends.

# Data Availability

All data in this manuscript are provided in the main and Supplementary files.

# Supplementary Information

# Acknowledgements

D Yang is currently funded by a Taiwan Cambridge Scholarship. AFJ Janssen and D Larrieu were funded by a Sir Henry Dale Fellowship jointly funded by the Wellcome Trust and the Royal Society (Grant Number 206242/Z/17/Z). AFJ Janssen was supported by a FEBS Long-Term Fellowship, a Leverhulme Trust Early Career Fellowship, and the Isaac Newton Trust. MO Ellis and the research in the Balmus laboratory is supported by the UK Dementia Research Institute, which receives contributions from UK DRI, the UK MRC, the Alzheimer's Society, and Alzheimer's Research UK. Research at TINS is supported by PNRR-III-C9-2022-I8.

## Author Contributions

D Yang: data curation, formal analysis, visualization, and writing—original draft.
A Lai: data curation, formal analysis, investigation, visualization, and writing—review and editing.
A Davies: data curation.
AFJ Janssen: supervision, investigation, and writing—review and editing.
MO Ellis: data curation.
D Larrieu: conceptualization, resources, supervision, funding acquisition, validation, investigation, project administration, and writing—review and editing.

## Conflict of Interest Statement

D Larrieu is an employee of Altos Labs. All other authors declare that they have no conflict of interest.

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
