## [Reviewer comments · Life Science Alliance]

Life Science Alliance

A Novel Role for CSA in Regulating Nuclear Envelope Integrity: Uncovering a Non-Canonical Function

Denny Yang, Austin Lai, Amelie Davies, Anne Janssen, Matthew Ellis, and Delphine Larrieu

DOI: <https://doi.org/10.26508/lsa.202402745>

Corresponding author(s): *Delphine Larrieu, University of Cambridge*

Review Timeline:	Submission Date:	2024-03-28
	Editorial Decision:	2024-03-28
	Revision Received:	2024-08-06
	Editorial Decision:	2024-08-08
	Revision Received:	2024-08-14
	Accepted:	2024-08-20

Transaction Report:

Please note that the manuscript was reviewed at Review Commons and these reports were taken into account in the decision-making process at Life Science Alliance.

March 28, 2024

Re: Life Science Alliance manuscript #LSA-2024-02745

Dr. Delphine Larrieu
Cambridge University
Pharmacology Department
University of Cambridge
Tennis Court Road
Cambridge, Cambridgeshire CB2 1PD
United Kingdom [GB]

Dear Dr. Larrieu,

Thank you for submitting your manuscript entitled "A Novel Role for CSA in the Regulation of Nuclear Envelope Integrity: Uncovering a Non-Canonical Function" to Life Science Alliance. We invite you to re-submit the manuscript, revised according to your Revision Plan.

Thank you for this interesting contribution to Life Science Alliance. We are looking forward to receiving your revised manuscript.

Sincerely,

B. MANUSCRIPT ORGANIZATION AND FORMATTING:

Revision Plan

Manuscript number: RC-2023-02303

Corresponding author(s): Delphine Larrieu

[The “revision plan” should delineate the revisions that authors intend to carry out in response to the points raised by the referees. It also provides the authors with the opportunity to explain their view of the paper and of the referee reports.]

The document is important for the editors of affiliate journals when they make a first decision on the transferred manuscript. It will also be useful to readers of the reprint and help them to obtain a balanced view of the paper.

*If you wish to submit a full revision, please use our "Full Revision" template. **It is important to use the appropriate template to clearly inform the editors of your intentions.**]*

1. Description of the revisions that have already been incorporated in the transferred manuscript

Please insert a point-by-point reply describing the revisions that were already carried out and included in the transferred manuscript. If no revisions have been carried out yet, please leave this section empty.

Reviewer #1

- 1- Albeit the link between CSA and NE integrity the work is in my eyes too preliminary. Although the data presented are well done and carefully evaluated they mostly (except Fig 1A) rely on direct comparisons of one patient cell line (CS-A or CS-B) to the same cells expression the wildtype protein. It remains thus open whether the effects seen on LEM2 expression, LEM2-LaimA/C interaction, stress fibre formation, cGAS/STING signaling pathway activation in the CS-A cells are representative for a number of different CS patient derived cells. This is especially important given the small changes observed. Please note that there are also clear differences between the CSA-wt and CSB-wt cells. Would the HPA CSA KO cells show in addition to NE irregularities (not even quantified) the same phenotypes and can they be reverted by re-expression of the wildtype CSA protein?

We would like to thank the reviewer for this comment. Indeed we previously observed nuclear circularity defects in CSA KO HAP1 cells but had not investigated the other phenotypes in this cell line. One of the main reasons behind this is the technical difficulties associated with performing immunofluorescence with HAP1 cells who are very small and tend to grow in aggregates.

To address this question, and in addition to using the CS-A patient derived cells, we have now added another Human fibroblast cell line (AG10803) in which we knocked out CSA using CRISPR/Cas9. Similarly to what we observed in the CS-A cells, we observed NE defects including abnormal nuclear shape, increased nuclear blebbing and cGas foci accumulation (Supplementary Figure 1). In addition, we observed the formation of actin stress fibers and decreased LEMD2 at the NE upon CSA KO (Supplementary Figure 3).

Revision Plan

- 2- The link between CSA and SUN1 is not well worked out. What is the effect of SUN2 downregulation and that of nespirins? It remains unclear whether the observed effects are indeed LINC mediated.

To address this point, we knocked down other components of the LINC complex including SUN2 and Nesprin1 in both CS-A and CSA KO cells as described above. Interestingly, we saw that SUN1 is the main mediator of these NE abnormalities, as depletion of the other LINC complex proteins could not rescue the NE defects including nuclear shape, nuclear blebbing, and cGAS foci formation. (Supplementary Figures S1 and S4).

Minor comments:

Fig 1B: Why is HA-Tagged CSA not shown on the CSA western? This would be helpful to compare to the endogenous levels at least in CSB cells. A western showing an housekeeping marker would allow better comparison. Judging from the proteins markers HA-tagged CSA seems much larger as endogenous CSA (first versus second row). Again, less cropped western blots would help.

We are sorry for the confusion and we have realised that the molecular weights on the western blots were incorrectly labeled on this figure. We have now modified this on the Figure.

Fig 3A: Is CSA FLAG or HA-tagged? Or both? If both are expressed the question raises of why the CSA-LEM 2 interaction is only seen in an overexpression situation.

CSA is HA tagged. We have attempted to perform the immunoprecipitation on endogenous proteins, but were not able to detect the interaction in these conditions (data not shown). This could be due to the difficulty to solubilize enough of the endogenous LEMD2 protein or to other technical difficulties that we could not resolve.

Fig 5: Inconsistency between figure and figure legend: 20 vs 25 nM Jasplakinolide. I assume Latrunculin A should read Cytochalasin D?

Thank you for pointing this out, and yes this is indeed a mistake on our labeling. This has been rectified on the figure legend.

Fig5B: Not clear why in "CSA-wT cells" Cytochalasin D and Jasplakinolide have the same effect on nuclear envelope shape yet only Jasplakinolide increases the number of blebs.

Cyt D inhibits actin polymerization while jasplakinolide increases polymerization. It is likely that actin polymerization increases blebs through exerting extra forces on the nucleus through actin cables/ actin based motility, therefore enhancing NE curvature and blebs.

Page 10: Method for IF: 4% (v/v) paraformaldehyde and 2% /v/v) should likely read (w/v).

Page 19: replace "withl" by "with".

We have modified both these points.

Reviewer #2

Revision Plan

The paper is well-written and for the most part, the data support the conclusions of the authors. Some minor caveats could be addressed to improve the quality of the manuscript.

We would like to thank the reviewer for their positive feedback on our manuscript.

- The phenotype of decreased LEMD2 incorporation into the NE in CS-A cells is minor. Only ~20% and thus, it is not clear whether this is causal of any of the NE abnormalities. It should be better explained how these data add to the story.

To address this point, we overexpressed LEMD2-GFP in CS-A cells and assessed nuclear envelope phenotypes (nuclear form factor and nuclear blebs).

We confirmed that LEMD2 overexpression could rescue these NE abnormalities (Figure 3C-E)

- Inducing actin polymerization and depolarization impact nuclear morphological abnormalities and nuclear blebbing. Do these treatments impact nuclear fragility and cGAS accumulation at NE break sites?

This is a good point indeed. To address this question, we assessed cGAS foci accumulation in cytochalasin D and jasplakinolide treated cells. Our results confirmed that destabilizing actin using cytochalasin D reduces cGAS foci number in CS-A cells. When we stabilize actin using jasplakinolide, this causes increased number of cGAS foci in WT(HA-CSA) cells but did not further increase the number of cGAS foci in CS-A cells (Figure 6F).

- Depletion of SUN1 in CS-A cells increased nuclear circularity, decreased blebbing, and phosphorylation of TBK1. The impact of SUN1 depletion in cGAS foci formation at NE break sites and phosphorylation of STING is not shown. Such experiments will provide stronger evidence that CS-A activates the cGAS-STING pathway in a SUN1 (mechanical stress)-dependent manner.

We have addressed this question by analysing cGAS foci and cGAS-STING pathway activation upon SUN1 depletion by siRNA. We showed that depleting SUN1 indeed reduces cGAS foci formation at the NE confirming that this process is SUN1-dependent (Figure 7H-I).

Reviewer #3

The data are generally clear, well performed and well interpreted with some exceptions:

- 1) I appreciate the use of isogenic cell lines (a big plus when dealing with patient-derived cell lines). However, these lines were established 30 years ago and the reported phenotypes might be due to genetic drifts. To exclude this, I suggest to complement the HAP-1 ERCC8 KO cell line with exogenously expressed CSA and assess if this rescues the phenotypes reported. Validation of the KO in these lines, either by western blotting or sequencing is needed.

We would like to thank the reviewer for this comment. We have now addressed this point as described in the response to reviewer 1 (see above)

Revision Plan

2) Related to the complementation of patient cell lines, the exogenous HA-CSA is not recognised by the anti-CSA in the CSA-null patient cell lines (Fig 1B, second blot). Shouldn't you be able to see this exogenous protein? HA-GFP-CSB in the complemented CSB-null patient cell line runs at the same weight as endogenous CSB (Fig 1B, fourth blot). This is also unexpected. I think you need better characterisation of your cell lines and need to demonstrate the level of exogenous transgenes that have been used to complement the cells and that they localise appropriately, presumably to the nucleus. You should also make sure to cite the paper where they were isolated and describe that they were immortalised (Troelstra et al., 1992) and the paper in which transgenes were stably overexpressed (Qiang et al., 2021).

As mentioned in the above response to reviewer 1, we have rectified some mistakes from the previous labeling of the molecular weight on this western blot. This will be corrected and the full uncropped western blot will be provided as a supplementary figure.

We have also now added the suggested references (Ref 40-41).

3) Immunolocalisation of INM proteins is notoriously tricky and the permeabilisation steps include only 0.2% Tx100, which can be insufficient to permeabilise the INM. I appreciate the Emerin and Lamin immunostaining seems to have worked, but in many cases successful immunostaining can be antibody-specific. Can you try harsher permeabilisation to expose LEM2 epitopes? I'm somewhat uncomfortable with the suggestion that there is a cytosolic (ER?) pool of endogenous LEM2 as this runs counter to the literature and feel that your antibody or fixation conditions are illuminating a non-specific protein. The WB in Fig 2E shows that there is virtually no LEM2 in the "soluble" fraction. I would be more cautious on this cytoplasmic/nuclear pool interpretation. Biochemical nuclear and cytoplasmic fractionation would help clarify the signal in a NE vs a non-NE pool.

As suggested by the reviewer, we tested a harsher permeabilization using 0.5% triton for the immunofluorescence staining of LEMD2. It did not appear any different compared to permeabilization using 0.2% triton, which was initially used throughout the manuscript. Therefore, we did not include this result in our manuscript. Below is an image showing LEMD2 staining using 0.5% triton permeabilization:

As we suggest in the manuscript however, we think that the "cytoplasmic" LEMD2 pool we observed by IF in the absence of pre-extraction is indeed unspecific. This is why we have performed the rest of the experiments with a pre-extraction step, that we have shown to give a specific LEMD2 signal that disappear upon depleting LEMD2 by siRNA.

Revision Plan

4) Pag 15: "Using a Proximity Ligation Assay (PLA), we showed a significant reduction in the number of PLA foci in CS-A cells compared to the WT(HA-CSA) cells, reflecting a reduced number of LEMD2-lamin A/C complexes (Figure 2G, 2H). This data suggests defects in the incorporation of LEMD2 into the NE and lamin protein complexes in CS-A cells". If you have less LEM2 in the NE, it is quite expected that you will have less "LEM2-laminA/C" complexes. To me the logic doesn't hold and this data does not suggest that there is an underlying defect in LEM2-lamin interaction. To ascertain whether there is such a defect one could perform an IP against LEM2 and quantify laminA/C, normalizing by the amount of LEM2 in the input.

We feel we may not have been clear in how we interpreted this data. What we mean is that in each individual cell, the number of Lamin-LEMD2 complexes is decreased, probably indeed due to the fact that there is less LEMD2 altogether within the nucleus in the absence of CSA. We have clarified this in the text.

5) "We overexpressed LEMD2-GFP and Flag-CSA constructs, followed by GFP pulldown in WT(HA-CSA) cells". Since the co-IP data are obtained in overexpression conditions (of both HA-CSA and

Revision Plan

Flag-CSA?), the authors should validate the interaction between LEM2 and CSA using an orthogonal approach. Perhaps anti-HA capture of the WT(HA-CSA) cells would allow you to immunoblot for endogenous LEM2?

As detailed above, we tried to perform the interaction experiment with endogenous proteins without success. This could be due to technical difficulty, including the insoluble properties of the endogenous LEMD2 protein.

6) Related to the CSA-LEM2 binding in the above experiment, the procedure involves combining a native detergent-extracted cytoplasmic pool with a denatured (RIPA-extracted) nuclear pool for performing the GFP-trap. From which pool was the tagged CSA bound to LEM2 in?

We are sorry about the confusion. We didn't try to run the IP from the different pools but instead from the combined pools, to ensure we were looking in the whole cell extract. We would expect however that the interaction occurs in the nuclear pool as both CSA and LEMD2 are nuclear proteins.

7) "The absence of CSA in CS-A patient cells does not affect the mobility of LEMD2 at the NE but instead decreases its interaction with A-type lamins". To me the fact that loss of CSA decreases LEM2-lamin interaction is not well supported (see point 3).

See our response to point 4

8) "Here, we showed by immunoprecipitation that LEMD2 also interacts with CSA. This suggests that the recruitment and stabilization of LEMD2 to the NE is mediated by an interaction with CSA, although the mechanism remains unclear". I think this is an overstatement: there are no data suggesting that CSA recruits or stabilises LEM2 at the NE.

We have now toned down this statement in the text: Here, we showed by immunoprecipitation that LEMD2 also interacts with CSA. This could suggest that the recruitment and stabilization of LEMD2 to the NE is mediated by an interaction with CSA, although the mechanism remains unclear and would require further experimentation.

9) As the authors suggest in the discussion, it would be worth checking whether LEM2 overexpression is able to rescue some of the NE defects reported, strengthening the hypothesis that LEM2 levels are at least in part responsible for the phenotypes reported.

To address this point, we overexpressed LEMD2-GFP in CS-A cells and assessed nuclear envelope phenotypes (nuclear form factor and nuclear blebs).

We confirmed that LEMD2 overexpression could rescue these NE abnormalities (Figure 3C-E)

10) To me it is not clear how the reported phenotypes are interrelated. The first part of the manuscript shows that CSA interacts with LEM2, and that loss-of-function CSA impacts on LEM2 levels and LEM2-lamin interaction, suggesting a direct role for CSA at the nuclear envelope. The second part of the manuscript shows that cells with defective CSA have more actin stress fibres and releasing the cytoskeleton-nuclear tethering is able per se to rescue the nuclear membrane and cGAS phenotypes. How do the authors reconcile these two parts? Is CSA directly involved in both inner nuclear membrane homeostasis and actin cytoskeleton modulation or is this latter role upstream and the NE defects a mere consequence of increased cytoskeleton rigidity?

At this point indeed we cannot draw definitive conclusions as to whether the two described phenotypes are inter-related. However, by addressing the other points raised by the reviewers, we hope this has helped clarifying the mechanism.

Revision Plan

11) It is not clear how or why actin stress fibres are elevated in the CS-A cells. Can the authors provide any insight based on their RNAseq analysis? Demonstrating a link to ROCK, LIMK or Rho signalling would be interesting and verifying ppMLC2 levels would help explain why contractility is enhanced.

Additionally, is the increase in contractility dependent upon any of the genes identified as up- or downregulated in RNAseq? Presently, the manuscript is missing a link between its two halves.

We would like to reiterate that the RNASeq analysis we performed was done on previously published data from another group (as described in the text). To address the point raised by the reviewer, we have looked more specifically into our analysis to look at ROCK, LIMK or Rho signalling but did not see modulation of these pathways in absence of CSA. We feel that establishing the link between the two phenotypes is outside the scope of this manuscript.

12) Related to point 1, the RNAseq comparison was performed on patient cells lacking CS-A and patient cells lacking CS-A and later over-expressing HA-CSA, and this comparison is used extensively for phenotype description in the manuscript. In isn't clear to me that this is the most insightful comparison to make; the rescue by overexpression is not as elegant as CRISPR reversion and the ko fibroblasts have presumably been surviving well in culture without CS-A before this protein was overexpressed. Can you validate the differential expression of any identified proteins in the acute HAP1 ko? Can you validate any of the differentially expressed proteins in comparison to normal fibroblasts (e.g., 13O6, as per Qiang et al., 2021)?

We have now validated most of the NE phenotypes observed in this work in another fibroblast cell line (AG10803) upon acute CSA KO by CRISPR/Cas9, including nuclear shape, nuclear blebs, cGAS foci, F-actin level, nuclear LEMD2 localization.

Our results showed that knocking out CSA in AG10803 cell line induces NE defects similar to the ones observed in the CS-A cell line including aberrant nuclear shape, increase in nuclear blebbing and cGAS foci. We also observed formation of actin stress fiber and mislocalization of LEMD2 at the NE in CSAKO AG10803 cells.

Minor comments

- Page 14: "To characterize the NE phenotypes further, we obtained CS patient-derived cell lines carrying loss-of-function mutations in CSA (CS-A cells) or CSB (CS-B cells), and their respective isogenic control cell lines (WT(HACSA) and WT(HA-GFP-CSB))." What type of loss-of-function? Is the mutant protein still produced? In Fig 6A there seem to be a band in the CS-A blot (second lane), but in Fig 1B, there isn't. I think this is important to know to interpret the phenotype related to LEM2 interaction.

We have clarified that in the text: we obtained CS patient-derived cell lines carrying loss-of-function mutations causing destabilization of CSA and CSB in CSA (CS-A cells) or CSB (CS-B cells). Indeed, the loss of function mutation leads to the absence of CSA protein.

- Figure 1B is poorly annotated. What do - and + stand for? In general, I find a bit confusing how the WB are presented throughout the manuscript, specifically how the antibodies are reported (e.g., HA-CSA instead of HA). Please mark up all western blots with antisera used. Please make sure all expected bands are within the crops - e.g., Fig 3B, the anti-LEM2 blot should be expanded vertically to show the LEM2-GFP relative to endogenous LEM2.

We have corrected these on the figures

Revision Plan

- From the methods, it appears that you obtained a LEM2-GFP, and cloned it into an expression vector (pEGFPC1) to make GFP-LEM2. Please provide clarity on which construct was used in which figure, and verify that an N-terminally tagged LEM2 still localises to the NE.

We actually cloned LEMD2 into an empty pEGFP vector but still maintained LEM2-GFP. We have removed the C1 plasmid from the methods to avoid confusion as we removed the MCS and GFP and just used the blank vector and inserted lem2-gfp as we obtained it.

- Fig 1I: there is some text on top of the upper panels (DAPI, cGAS, Merge).

Revision Plan

"Through gene ontology analysis, we found that genes involved in endoplasmic reticulum (ER) stress were differentially expressed (Figure 4B)". I don't think that the way data are shown in Fig 4B is effective. Since GO has been performed, I would replace the table with a GO enrichment analysis graph. Ensure to report all the data in a supplementary .xls so that others can see and reuse it. Is there a mandated repository that accepts RNAseq data?

The RNAseq experiment and data was performed by another group and reported in a previous study, as referenced in the main text of the manuscript (*Epanchintsev A, Costanzo F, Rauschendorf MA, Caputo M, Ye T, Donnio LM, et al. Cockayne's Syndrome A and B Proteins Regulate Transcription Arrest after Genotoxic Stress by Promoting ATF3 Degradation. Mol Cell. 2017 Dec;68(6):1054-1066.e6.*). Here, we only re-analysed their data using STRING pathway analysis, as detailed in the Material and Methods. However, as suggested by the reviewer, we have replaced the table by a GO enrichment graph.

We included GO enrichment analysis using DE genes from the RNA-seq analysis comparing CS-A and WT cells as supplementary excel files.

- The volcano plot looks weird with many values at the maximum log10 (P-value) - is the data processed appropriately?

As mentioned above, the RNA Seq analysis was performed and published in a different study. We think this is because the Y axis shows adjusted P values.

- Figure 5B: the legend says "Latrunculin A". Please correct.

This has been corrected

- For a Wellcome funded researcher, I'm surprised that the mandated OA statement and RRS is absent from the acknowledgements.

We of course want to comply with the open access policy of the Wellcome Trust. However, and based on the WT requirements detailed on their website, we believe the acknowledgement section complies with the funder: "All research publications must acknowledge Wellcome's support and list the grant reference number which funded the research reported."

2. Description of analyses that authors prefer not to carry out

Please include a point-by-point response explaining why some of the requested data or additional analyses might not be necessary or cannot be provided within the scope of a revision.

Revision Plan

This can be due to time or resource limitations or in case of disagreement about the necessity of such additional data given the scope of the study. Please leave empty if not applicable.

- We did not see up/downregulation in ROCK, LIMK, or Rho signaling pathway altered in the RNA-Seq data, cytoskeleton changes in CS-A cells are probably regulated through an unknown CSA-dependent process.
- We did not quantify cGAS foci in LEMD2-GFP infected cells, because transiently introducing DNA leads to cGAS/STING activation, which may consequently skew our interpretation. We did however quantify other NE phenotypes including nuclear roundness and nuclear blebbing which showed significant rescue in CS-A cells when LEMD2-GFP is overexpressed.
- We tried to immunoprecipitate HA-CSA and look at endogenous LEMD2, but we could only detect the interaction in a CSA and LEMD2 overexpression model.

August 8, 2024

RE: Life Science Alliance Manuscript #LSA-2024-02745R

Dr. Delphine Larrieu
University of Cambridge
Pharmacology Department
Tennis Court Road
Cambridge, Cambridgeshire CB2 1PD
United Kingdom

Dear Dr. Larrieu,

Thank you for submitting your revised manuscript entitled "A Novel Role for CSA in Regulating Nuclear Envelope Integrity: Uncovering a Non-Canonical Function". We would be happy to publish your paper in Life Science Alliance pending final revisions necessary to meet our formatting guidelines.

- please be sure that the authorship listing and order is correct
- please upload all figure files as individual ones, including the supplementary figure files; all figure legends should only appear in the main manuscript file
- please upload your main manuscript text as an editable doc file
- please add a Summary Blurb/Alternate Abstract to our system
- please add the Twitter handle of your host institute/organization as well as your own or/and one of the authors in our system
- titles in the system and manuscript file should match
- please consult our manuscript preparation guidelines <https://www.life-science-alliance.org/manuscript-prep> and make sure your manuscript sections are in the correct order
- please upload your Tables in editable .doc or excel format. They can be included at the bottom of the manuscript file or sent separately
- please add an Author Contributions section to your main manuscript text
- please add your main, supplementary figure, and table legends to the main manuscript text after the references section

FIGURE CHECKS:

- please add scale bars to Figure 2C, 4B, and 6A

A. FINAL FILES:

-- Summary blurb (enter in submission system): A short text summarizing in a single sentence the study (max. 200 characters including spaces). This text is used in conjunction with the titles of papers, hence should be informative and complementary to

the title. It should describe the context and significance of the findings for a general readership; it should be written in the present tense and refer to the work in the third person. Author names should not be mentioned.

B. MANUSCRIPT ORGANIZATION AND FORMATTING:

Sincerely,

August 20, 2024

RE: Life Science Alliance Manuscript #LSA-2024-02745RR

Dr. Delphine Larrieu
University of Cambridge
Pharmacology Department
Tennis Court Road
Cambridge, Cambridgeshire CB2 1PD
United Kingdom

Dear Dr. Larrieu,

Thank you for submitting your Research Article entitled "A Novel Role for CSA in Regulating Nuclear Envelope Integrity: Uncovering a Non-Canonical Function". It is a pleasure to let you know that your manuscript is now accepted for publication in Life Science Alliance. Congratulations on this interesting work.

DISTRIBUTION OF MATERIALS:

Again, congratulations on a very nice paper. I hope you found the review process to be constructive and are pleased with how the manuscript was handled editorially. We look forward to future exciting submissions from your lab.

Sincerely,
